# Fast and robust analog in-memory deep neural network training

**Malte J. Rasch** [1,2] ✉, **Fabio Carta**[1], **Omobayode Fagbohungbe**[1] & **Tayfun Gokmen** [1] ✉

Analog in-memory computing is a promising future technology for efficiently accelerating deep learning networks. While using in-memory computing to accelerate the inference phase has been studied extensively, accelerating the training phase has received less attention, despite its arguably much larger compute demand to accelerate. While some analog in-memory training algorithms have been suggested, they either invoke significant amount of auxiliary digital compute—accumulating the gradient in digital floating point precision, limiting the potential speed-up—or suffer from the need for near perfectly programming reference conductance values to establish an algorithmic zero point. Here, we propose two improved algorithms for in-memory training, that retain the same fast runtime complexity while resolving the requirement of a precise zero point. We further investigate the limits of the algorithms in terms of conductance noise, symmetry, retention, and endurance which narrow down possible device material choices adequate for fast and robust in-memory deep neural network training.

Analog in-memory computing AIMC is a promising future hardware technology for accelerating deep-learning workloads. Great energy efficiency is achieved by representing weight matrices in resistive elements of crossbar arrays and using basic physical laws of electrostatics (Kirchhoff's and Ohm's laws) to compute ubiquitous matrix-vector multiplications (MVMs) directly in memory in essentially constant time $\mathcal{O}(1)$[1–5]. Many recent AIMC prototype chip-building efforts to date have been focused on accelerating the inference phase of deep neural networks (DNNs) trained in digital[6–12]. However, in terms of compute requirements, the training phase is typically orders of magnitude more expensive than the inference phase, and thus would in principle have a greater need for efficient hardware acceleration using in-memory compute[13]. However, accelerating the training phase using AIMC has been challenging, in particular, because of the asymmetric and non-ideal switching of the memory devices that fail to achieve the high precision requirements of standard (SGD) algorithms designed for FP (FP) DNN training (see e.g., ref. 14 for a discussion). Thus, dedicated AIMC training algorithms are needed that can successfully train DNNs with the promised AIMC speedup and efficiency despite non-ideal device switching characteristics.

To accelerate DNN training in contrast to inference, the back-propagation of the gradients in SGD, as well as weight gradient computation and weight update itself, have to considered. While the backward pass of an MVM is straightforwardly accelerated in AIMC by transposing the inputs and outputs in constant time $\mathcal{O}(1)$, the gradient accumulation and update onto weights represented in the conductances of the memory elements is much more challenging. Typical device materials, such as Resistive Random Access Memory (ReRAM)[15], Electro-Chemical Random Access Memory (ECRAM)[16,17], as well as capacitors as weight elements[18], show various degrees of asymmetry when updating the conductance in one direction versus the other direction, as well as a gradual saturation to a minimal or maximal conductance value. Moreover, the device conductance can only efficiently be updated in small increments thus making some operations such as a full reset to a common target conductance prohibitively expensive. Finally, inherent device-to-device variations make it

[1]IBM Research, TJ Watson Research Center, Yorktown Heights, NY, USA. [2]Sony AI, Zürich, Switzerland. ✉e-mail: malte.rasch@gmail.com; tgokmen@us.ibm.com

challenging to implement many algorithmic ideas that instead inherently assume translational invariance.

One way to get around these challenges is to sacrifice speed and efficiency by simply computing the gradient and its accumulation in digital memory and precision and only accelerate the forward and backward pass using AIMC, as suggested by Nandakumar et al.[19,20]. However, given that $\mathcal{O}(N^2)$ digital operations are needed for updating a weight matrix of size $N \times N$, the update phase would not match with the $\mathcal{O}(1)$ character of the MVM in the forward and backward passes and thus slow down the overall AIMC acceleration of DNN training.

Therefore, Gokmen et al.[13] instead suggested to use coincidence of voltage pulse trains to perform the outer-product and weight update operations fully in-memory in a highly efficient and fully parallel manner. This approach has great potential since also the update phase can then be done in constant time $\mathcal{O}(1)$. Unfortunately, when computing the gradient and directly updating the weight in-memory with this approach, a bi-directionally switching device of unrealistically high symmetry and precision is needed[13,21,22]. The main problem when accumulating gradients over time using asymmetric devices with realistic device-to-device variations is that each device will drift in general towards a different conductance value even in the case when random fluctuations with zero mean are accumulated and therefore the net update should be zero and identical for all devices.

However, realizing this issue, follow-up studies[23,24] more recently suggested to use two additional, separate arrays of non-volatile memory (NVM) devices to, respectively, accumulate the gradients separately from the weights and represent predetermined reference values. It turns out that a differential read of the devices used for the accumulated gradients and those programmed with the reference values can statically correct for the effect of the device-to-device variations on the gradient accumulation. Indeed, when additionally introducing a low-pass digital filtering stage, the requirements of the number of reliable conductance states and on-device symmetry were considerably relaxed[24]. Furthermore, because only $\mathcal{O}(N)$ additional digital operations are needed, the update pass retains very good runtime complexity and is this efficiently accelerated using AIMC.

While this Tiki-Taka version 2 (TTv2) algorithm[24] was also demonstrated recently in hardware and in simulation using realistic ReRAM on small tasks[25], several challenges remain in practice. First, implementing the circuitry for a differential read results in a more complicated unit cell design as well as significant additional chip area cost for the additional reference devices. Second, the estimation of the reference conductance values and the programming of the resulting values has to be done prior to the start of the training, which takes additional time and effort[26]. Finally and most importantly, as we will show here, even a small deviation of the programmed reference values from the theoretical values on the order a few percent leads to significant accuracy drops during training, thus severely limiting this approach in practice where much larger programming errors and limited retention are common issues. Indeed, even in the study demonstrating the TTv2 algorithm[25], reference values were represented in digital values due to test hardware limitations. Moreover, even if the programming would be perfect, retention of the exact values over long training times might become problematic. Together, these issues make the use of the TTv2 algorithm challenging in practice.

Here, we first make a simple improvement to the TTv2 algorithm to better handle any offsets inflicted by an erroneous reference value. We propose to use the chopper technique[27] in the gradient accumulation to remove any remaining offsets in the reference by periodic or random sign changes. This Chopped-TTv2 (c-TTv2) algorithm relaxes the requirement of the reference errors to smaller than about 25% without significantly altering the runtime in comparison to TTv2. Secondly, we introduce an altogether different algorithm, Analog Gradient Accumulation with Dynamic reference (AGAD), that establishes reference values on-the-fly using a modest amount of additional digital computing. In this case, the reference values are an estimate of the recent past of the transient conductance dynamics and thus independent of any device measurement or device model assumption. We find that both c-TTv2 and AGAD train benchmark DNNs to state-of-the-art accuracy. In addition, AGAD also greatly simplifies the hardware design as it does not need a separate conductance array for any reference values, nor any differential read circuitry. We also show that AGAD broadens the choice of device materials since both symmetric as well as asymmetric device characteristics can be used, in contrast to TTv2 and c-TTv2, which depend on devices showing asymmetry. By estimating the expected performance, we show that the both proposed algorithms retain the fast runtime of TTv2, showing two orders of magnitude runtime improvement to the alternative approach using digital instead of in-memory gradient accumulation[20].

Finally, we also introduce a dynamic way to set the learning rate to optimize the gradient accumulations in diverse DNNs, significantly easing the search for hyper-parameters in practice.

## Results

In the following, we present first simple toy examples to illustrate and compare the mechanism of the proposed training algorithms Chopped-TTv2 (c-TTv2) (Supplementary Alg. 2) and Analog Gradient Accumulation with Dynamic reference (AGAD) (Supplementary Alg. 3) to the baseline Tiki-Taka version 2 (TTv2) algorithm (see Fig. 1; the proposed algorithms are described in detail in the "Methods" section "Fast and robust in-memory training"). Then, we use them to simulate the training of DNNs with different material and reference offsets settings. For simulations, we use the PyTorch-based[28] open source toolkit (AIHWKit)[29], where we have implemented the proposed algorithms (see also Supplementary Fig. 4). Finally, we investigate the projected performance numbers, as well as on-chip memory, and digital compute, and device material requirements.

### Gradient update mechanisms

All here proposed AIMC learning algorithms share the feature that they use a dedicated array of conductances (that is $\breve{A}$) to compute the gradient accumulation in-memory, while slowly transferring the accumulated gradients onto the actual weight matrix, which is represented by another crossbar array of conductances (that is $\breve{W}$) to enable in-memory acceleration of the forward and backward passes as well. To illustrate the mechanism of the proposed learning algorithms, we first investigate a simple case where activations are given by $x = -X$ and gradient inputs by $d = \alpha X + (1 - \alpha)Y$ where $X, Y \sim \mathcal{N}(0, 1)$ are Gaussian random variables. Thus, in this case, the correlation of activations and gradients is given by $\alpha$ and expected average update is only in one direction $\Delta \breve{w} \propto -\alpha$.

Let's first assume that the reference matrix $\breve{R}$ used for the differential read of the accumulated gradients in TTv2 and c-TTv2 (see Fig. 1) is perfectly accurately set to the symmetry point (SP) of $\breve{A}$ (as illustrated in Fig. 2) so that no offset remains (see results in Fig. 3A–C). For simplicity, we plot here the conductance values in normalized units, assuming that the SP is set arbitrarily to zero, $\breve{a}^* \equiv 0$, and the maximal and minimal conductance at 1 and −1, respectively (see "Methods" section "Device material model" for details). Note that for TTv2 (Fig. 3A; see "Methods" section "Recap of the Tiki-Taka (version 2) algorithm") the trace of a selected matrix element $\breve{a}$ is strongly biased towards negative values, thus indicating correctly the direction of the gradient. It, however, saturates at a certain level, caused by the characteristics of the underlying device model (see Eq. (4)). Because of the occasional reads (indicated with dot markers), the hidden weight accumulates until threshold is reached at −1 (green trace), in which case the weight $\breve{w}$ is updated by one pulse (orange trace). The shaded blue area indicates the instantaneous accumulated gradient value of

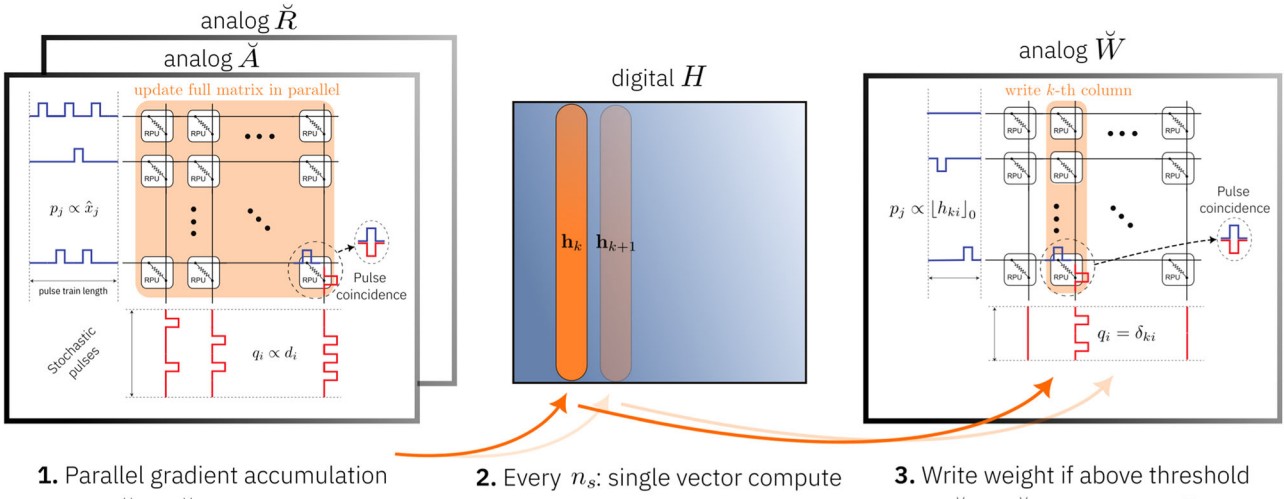

**Fig. 1 | Illustration of gradient update computation steps.** The general structure of the gradient computation is shared for all improved learning algorithms discussed and is based on Tiki-Taka version 2 (TTv2) (see ref. 24). For each input vector **x** and backpropagated error vector **d** the weight gradient is first accumulated on a crossbar array $\breve{A}$, using a parallel pulsed outer-product update with learning rate $\lambda_A$ ([13]; see Supplementary Alg. 1). Note that the matrices are here displayed in a transposed fashion so that voltage inputs **x** are delivered from the left and **d** from the bottom side. Then a single row of the accumulated gradient in $\breve{A}$ is read out intermittently every $n_s$ vector updates (looping through the rows over time), and digital computation is used to arrive at a FP vector $\mathbf{z}_k$ that is added to the digital storage $H$ with learning rate $\lambda_H$. Finally, the corresponding row of actual weight matrix, which is represented by a second crossbar array $\breve{W}$, is updated when a threshold is crossed, and the hidden matrix $H$ is reset correspondingly. The newly proposed algorithms differ in the digital computation to arrive at $\hat{\mathbf{x}}$ and $\mathbf{z}_k$. For the TTv2 baseline algorithm, it is $\hat{\mathbf{x}} \equiv \mathbf{x}$ and $\mathbf{z}_k \equiv (\breve{A} - \breve{R})\mathbf{v}_k$ where the reference crossbar array $\breve{R}$ is programmed before DNN training and a fast differential analog MVM is used for readout (using one-hot unit vector $\mathbf{v}_k$). See "Methods" section "Fast and robust in-memory training" and Supplementary Fig. 2 for more details on the digital operations of the proposed algorithms.

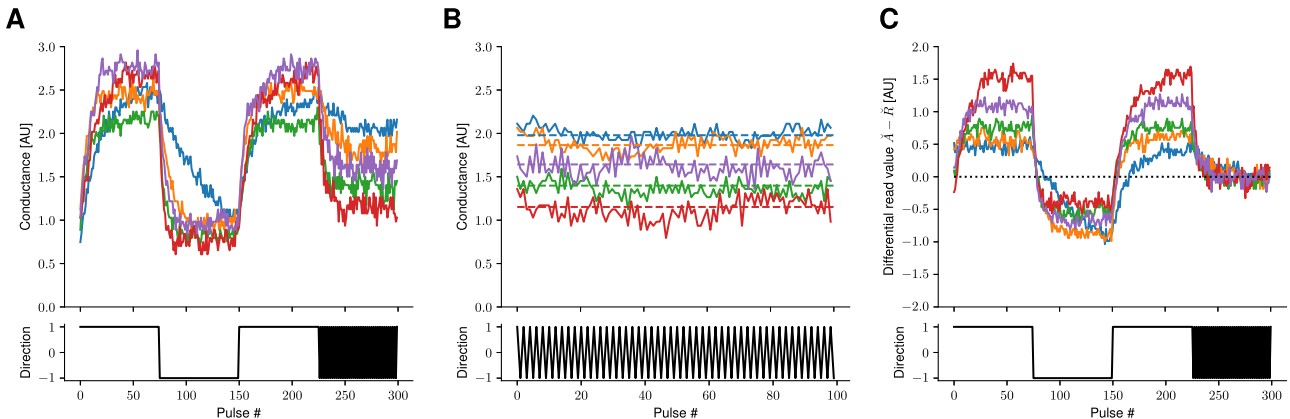

**Fig. 2 | Setting the zero-reference point with differential read in Tiki-Taka version 2 (TTv2) and Chopped-TTv2 (c-TTv2).** **A** Example ReRAM-like device response traces showing noise and variation in response to bi-directional pulses. Here we assume that the device gradually saturates with consecutive up or down pulses (see lower plot for pulse direction applied). Noise properties and update step sizes can be adjusted in the soft-bounds model Eq. (4) to e.g., reflect typical ReRAM (high noise), capacitor (medium noise, lower variation), or ECRAM (low noise) traces. **B** Due to the asymmetry, consecutive (pairwise) up-down pulses converge the conductance to a fixed point where up and down pulses are on average of the same size (symmetry point (SP), see Eq. (8)). Because of device-to-device variation each device has an individual SP value (dashed lines). **C** When the SP is estimated for each device of a crossbar array $\breve{A}$, it can be programmed on a separate reference device $\breve{R}$. Assuming that the circuitry allows for a matrix-vector multiplication with differential read, e.g., $y_i = \sum_j \left( \breve{a}_{ij} - \breve{r}_{ij} \right) x_j$, then individual device responses are effectively set to zero when consecutive up-down pair pulses are applied.

$\omega = \breve{a} - \breve{r}$. The area would be red if the value was positive, which would cause the hidden weight $h$ to update in the wrong direction if readout at that moment.

In Fig. 3B, the behavior of the proposed c-TTv2 algorithm (see "Methods" section "Chopped-TTv2 algorithm" for details) is shown for the same inputs. In this algorithm, the gradients are accumulated with changing signs in either positive or negative directions within a chopper period. Here, for better illustration, a fixed chopper period is chosen (gray dashed lines). Since the incoming gradient is constant (negative), the modulation with the chopper sign causes an oscillation in the accumulation of the gradient on $\breve{a}$. However, since the sign is corrected for during readout, the hidden matrix is updated (mostly) in the correct direction (blue areas are sign corrected). As we will see below, this flipping of signs will cancel out any offsets (which are currently assumed to be 0). If the trace of $\breve{a}$ has not returned to the SP before the readouts, it would cause some transient updates of the hidden weights in the wrong direction (red areas). The weight $\breve{w}$ is nevertheless correctly updated on average as the hidden weight

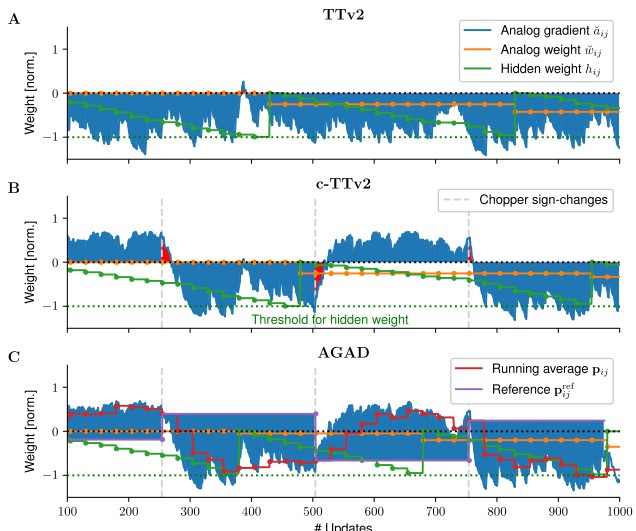

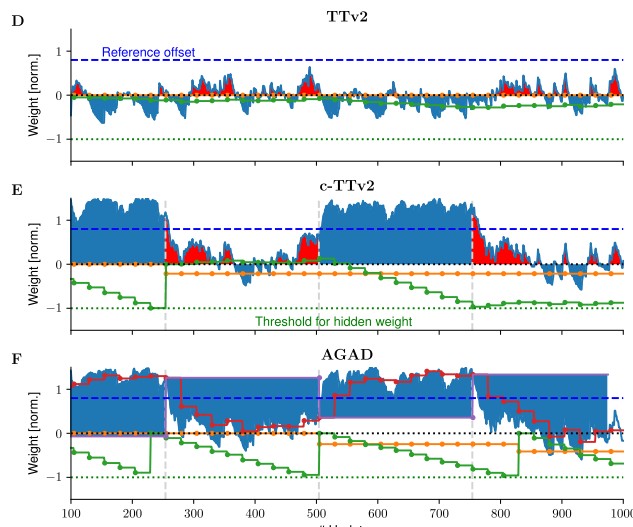

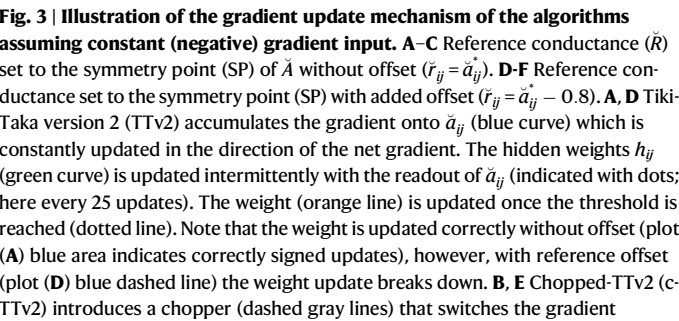

**Fig. 3 | Illustration of the gradient update mechanism of the algorithms assuming constant (negative) gradient input. A**−**C** Reference conductance ($\check{R}$) set to the symmetry point (SP) of $\check{A}$ without offset ($\check{r}_{ij} = \check{a}_{ij}^*$). **D**−**F** Reference conductance set to the symmetry point (SP) with added offset ($\check{r}_{ij} = \check{a}_{ij}^* - 0.8$). **A, D** Tiki-Taka version 2 (TTv2) accumulates the gradient onto $\check{a}_{ij}$ (blue curve) which is constantly updated in the direction of the net gradient. The hidden weights $h_{ij}$ (green curve) is updated intermittently with the readout of $\check{a}_{ij}$ (indicated with dots; here every 25 updates). The weight (orange line) is updated once the threshold is reached (dotted line). Note that the weight is updated correctly without offset (plot (**A**) blue area indicates correctly signed updates), however, with reference offset (plot (**D**) blue dashed line) the weight update breaks down. **B, E** Chopped-TTv2 (c-TTv2) introduces a chopper (dashed gray lines) that switches the gradient

accumulation direction (here set to regular intervals). Note that weight is correctly updated without offset (plot (**B**)), however, a reference offset (plot (**E**)) causes a slowdown (but not breakdown) of the weight learning as the offset disturbs the zero point in one chopper cycle but recovers every other cycle (red areas indicate wrong sign of the gradient readout due to the offset). **C, F** (AGAD) introduces an on-the-fly reference estimation ($p_{ij}$; red line) that is copied to the current reference ($p_{ij}^{ref}$, violet line) when the chopper changes. Note that in this case the reference is dynamically adjusted so that weight update is correct without (plot (**C**)) as well as with any offset (plot (**F**)). Parameter settings: $5 \times 5$ matrix size (only first element is plotted), $\delta = 0.05$, $\sigma_b = \sigma_\pm = \sigma_{\text{d-to-d}} = \sigma_{\text{c-to-c}} = 0.3$, $\gamma_0 = 200$, $\lambda = 0.1$, $n_s = 5$, $\beta = 0.5$, $\rho = 0.1$, $l_{\max} = 5$, $\lambda_A = 1$, and $\sigma_r = 0$.

averages out transients successfully. The rate of change of $\check{w}$, however, is somewhat impacted by the averaging of the transients.

We further propose the AGAD algorithm (Fig. 3C; see "Methods" section "AGAD algorithm" for details) that uses an (average) value $p^{ref}$ of the recent accumulated gradient $\check{a}$ as the reference point (and not the static SP programmed onto $\check{R}$; see violet line in Fig. 3C). The digital reference value of $p^{ref}$ is changed only when the chopper sign changes (dashed horizontal lines) and is computed by a leaky average of the past conductance readouts ($p$; see red line in Fig. 3C). Because of this on-the-fly reference computation, this algorithm is not plagued with the same transients. In fact, the increased dynamical range causes a faster update of the hidden matrix and subsequently the weight $\check{w}$.

Since in Fig. 3A the reference $\check{R}$ was set exactly to the SP of $\check{A}$−as required for TTv2−the zero point was perfectly set to the fix-point of the device dynamics[30]. In this case, the baseline algorithm TTv2 indeed works perfectly fine and might be the algorithm of choice, because it requires least amount of digital computing (as we discuss below). However, in a more realistic setting when the reference matrix $\check{R}$ does not exactly match the SP, that is programmed instead with an error offset $\check{R} \leftarrow \check{a}^* - \mu_r$ and $\mu_r \neq 0$, the algorithm performs generally poorly. This is shown in Fig. 3D, where the experiment of Fig. 3A is repeated, however, now with an offset of $\mu_r = -0.8$ (blue dashed line in Fig. 3D). Note that the constant gradient pushes the accumulated gradient $\check{a}$ away from the SP (here at zero) as expected, however, since the algorithm does subtract the offset programmed on $\check{R}$, the update onto the hidden matrix is wrong. In fact, hidden weight $h$ (green line) never reaches the threshold and is net zero in this example instead of becoming negative as expected (compare to the Fig. 3A).

On the other hand, because of the effect of the chopper sign changes, even this large offset can be successfully removed with the c-TTv2 algorithm (Fig. 3E). Note that the hidden weight $h$ as well as the weight $\check{w}$ decreases correctly. However, due to the large offset, noticeable oscillations (red areas) are stressing the accumulation on $h$,

thus reducing the speed and fidelity of the gradient accumulation. In case of the AGAD algorithm (Fig. 3 F), the dynamic reference point computation perfectly compensates any wrong offset, making the reference device conductance and the programming of the SP altogether unnecessary.

## Stochastic gradient descent on single linear layer
While investigating the case of constant gradient input is illustrative for the accumulation behavior of the learning algorithms, in a more realistic setting, the incoming gradient magnitude typically depends on the past update of the weight matrix, thus closing a feedback loop[30]. Therefore, we next test how the algorithms perform when actually implementing stochastic gradient descent. We first consider training to program a linear layer with output $f_i(\mathbf{x}) = \sum_{j=1}^{n} w_{ij} x_j$ to a given target weight matrix $\hat{W}$. We define the loss function as the mean squared deviation from the expected output by using the target weight $\hat{W}$, namely

$$L(\mathbf{x}|W, \hat{W}) = \frac{1}{2m} \sum_{i}^{m} \left( f_i(\mathbf{x}) - \sum_{j=1}^{n} \hat{w}_{ij} x_j \right)^2. \quad (1)$$

Naturally, when minimizing this loss (using SGD) and updating $W$, the deviation is minimized for $W = \hat{W}$. This problem statement is similar to the proposal to program target weights for AIMC inference[31], however, we here use our proposed gradient update algorithms to perform the gradient accumulation in memory instead of using digital computed gradients.

We set $\hat{W}$ to random values $\mathcal{N}(0, 0.3)$ and use $x_j \sim \mathcal{N}(0, 1)$ as inputs. We evaluate the different algorithms by the achieved weight error $\epsilon_w^2 = \langle (w_{ij} - \hat{w}_{ij})^2 \rangle$, that is the standard deviation (SD) of the learned weights with the target weight. Figure 4 shows the results for a $20 \times 20$ weight matrix after a set amount of inputs with fixed learning rate.

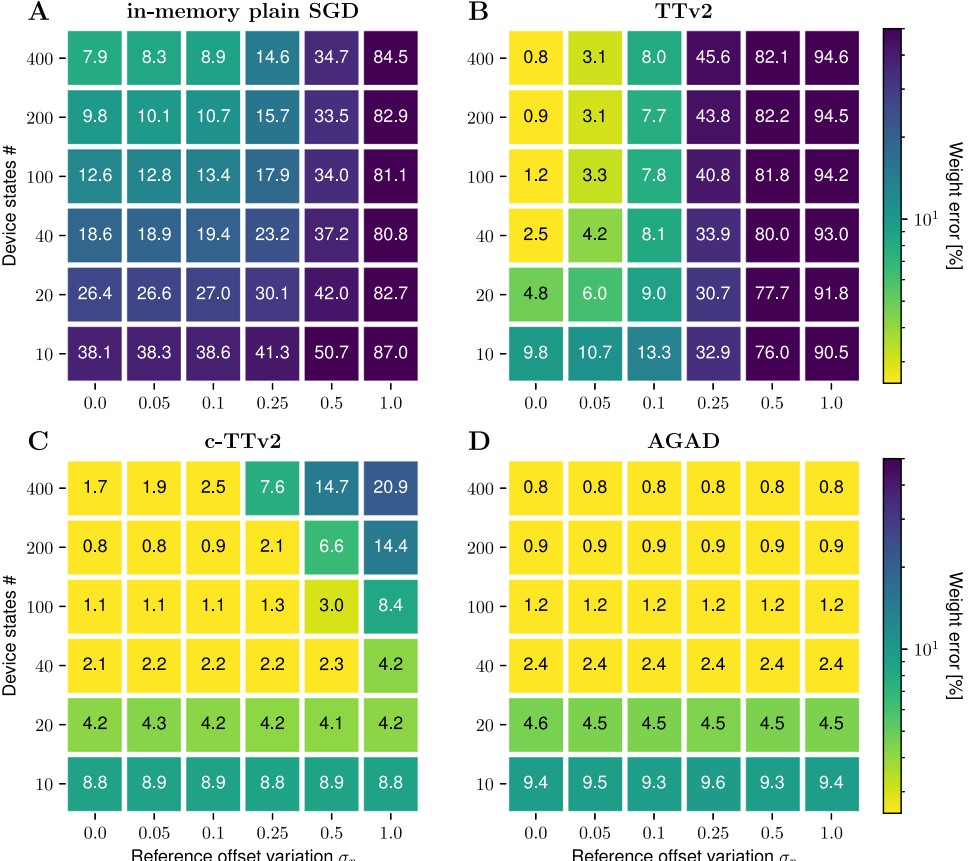

**Fig. 4 | Weight programming error using different learning algorithms.** Standard deviation of the converged analog weights $\check{W}$ to the target weights is plotted in color code. The value of the reference offset device-to-device variation $\sigma_r$ is increased horizontally, while vertically changing the number of material device states $n_{states}$ (see Eq. (6)). Less number of states, in general, corresponds to a noisier conductance response (e.g., for typical ReRAM materials), higher number of states corresponding to more ideal device conductance responses (e.g., ECRAM). **A** In-memory SGD using stochastic pulse trains. **B** Baseline Tiki-Taka version 2 (TTv2).

**C** The proposed Chopped-TTv2 (c-TTv2) algorithm. **D** The proposed (AGAD) algorithm. Simulation details: Parameter settings as in Fig. 3 except that $\sigma_r$ and $\delta$ are varied. Additionally, we set $\sigma_b = 0$ for $\check{W}$ only (to not confound results for not being able to represent the target weight with $\check{W}$) and set $\sigma_\pm = 0.1$ (to avoid a large impact of few failed devices on the weight error). The target matrix and inputs are fixed for each case for better comparison. Averaged over three construction seeds of the device variations.

We compare the two proposed algorithms (c-TTv2, and AGAD) with the TTv2 baseline[24], as well as with plain in-memory SGD, where the gradient update is directly done on the weight $\check{W}$ (Supplementary Alg. 1). Additionally, we explore the resilience to two-parameter variations, (1) the magnitude of the offset (by varying the SD of the reference offset $\sigma_r$ across devices), and (2) the number of device states $n_{states}$ (see Eq. (6)). As the number of states also scales the relative amount of conductance noise in our model (see "Methods" section "Device material model"), this variable can be seen as a choice of different device materials, where a low number of states corresponds to e.g., ReRAM devices, and a high number of states corresponds to e.g., ECRAM devices.

As expected in case of no offset $\sigma_r = 0$ and in agreement with the original study[24], the TTv2 algorithm works very well, vastly outperforming in-memory SGD, in particular for small number of states (e.g., $\epsilon_w \approx 5\%$ vs > 25.0%, respectively, for 20 states and the very same target weight matrix; see Fig. 4A, B). However, reference offset variations $\sigma_r > 0$ critically affect the performance of TTv2. As soon as $\sigma_r \geq 0.1$ (here corresponding to 5% of the weight range of 2), weight errors increase significantly (e.g., to $\epsilon_w \approx 9\%$ for 20 states). This poses challenges to the usefulness of TTv2 with current device materials because weight programming errors are generally in the order of at least 5–10% of the target conductance range for ReRAM ([6], see also Supplementary Fig. 1B in ref. 32). Thus, the reference $\check{R}$ cannot be programmed

accurately enough with the SP of $\check{A}$ (see "Methods" section "Recap of the Tiki-Taka (version 2) algorithm") to avoid a significant accuracy degradation when training in-memory using the baseline TTv2.

Using the concept of choppers in the proposed algorithms c-TTv2 and AGAD, on the other hand, improves the resiliency to offsets dramatically (Fig. 4C, D). The c-TTv2 algorithm maintains the same weight error for large offsets when the number of states is small. Offsets in case of larger number of states are less well corrected, consistent with the existence of transient decays towards the SP that are the slower the higher the number of states is (see Eq. (7)). In case of AGAD, reference offsets simply do not matter, as the reference is dynamically computed on-the-fly (see Fig. 4D). Moreover, in contrast to c-TTv2, AGAD works equally well for higher number of states showing that transients are not problematic here either.

## DNN training experiments

Finally, we compare the different learning algorithms for actual DNN training. For better comparison, we use largely the same DNNs that were previously used to evaluate the earlier algorithms. These were a three-layer fully connected DNN[13], LeNet convnet[33] for image classification of the MNIST dataset[34], and a two-layer recurrent long short-term memory (LSTM) network for text prediction of the War and Peace novel[24,35]. We again trained the DNNs with different reference offset variations (see Fig. 5; see Supplementary Methods Sec. C.1 for details)

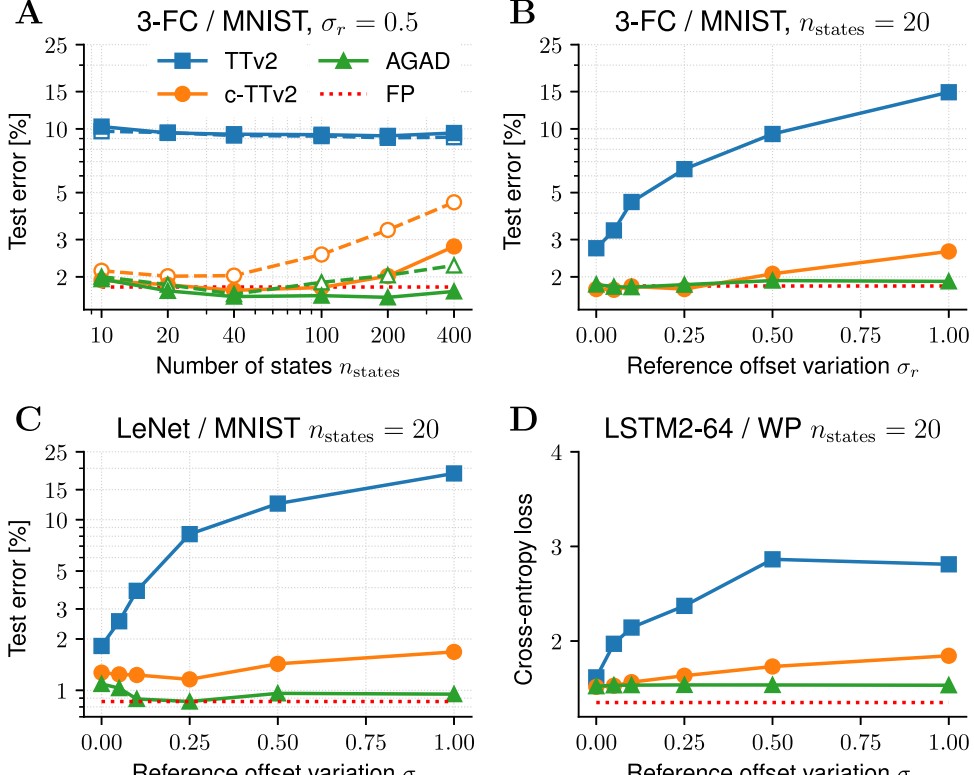

**Fig. 5 | DNN training with different analog learning algorithms.** The symmetry point (SP) of $\check{W}$ is either corrected for (closed symbols; compare to Fig. 2) or not (open symbols with dashed lines). All simulations are done for a fixed amount of epochs for comparability (see Supplementary Fig. 5 for example traces). **A** Converged test error (in percent) with a three-layer fully connected DNN on the MNIST dataset is shown for varying number of device states $n_{\text{states}}$ using a large reference offset variation $\sigma_r = 0.5$. Note that the proposed algorithms, Chopped-TTv2 (c-TTv2) and (AGAD), greatly outperform the baseline Tiki-Taka version 2 (TTv2) across all settings of $n_{\text{states}}$. Test errors for training in FP precision using standard SGD are shown as comparison (FP; red dotted line). **B–D** Reference offset variation $\sigma_r$ versus test error with $n_{\text{states}} = 20$ for different DNNs. Note that independent of the DNN the results are very similar to the weight programming task in Fig. 4: TTv2 essentially does not allow for reference offsets, TTv2 is much more tolerable, whereas AGAD is invariant against reference offset. **A, B** 3-layer fully connected DNN on MNIST. **C** LeNet on MNIST. **D** 2-layer LSTM on the War & Peace dataset. See Supplementary Methods Sec. C.1 for more details on the simulations.

with the same challenging device model (see example device response traces for $n_{\text{states}} = 20$ in Supplementary Fig. 3). As suggested by Gokmen[24], accuracy for all algorithms could in principle be further improved and weights could be extracted from the analog devices for further deployment using stochastic weight averaging not considered here.

The results of Fig. 5 are very consistent across the three different DNNs of various topologies (fully connected, convnet, and recurrent network) and confirm the trends found in case of the weight programming of one layer (compare to Fig. 4): If the offsets are perfectly corrected for, all algorithms fare very similarly reaching close to FP accuracy. However, as expected, the impact of a reference offset is quite dramatic for TTv2, whereas c-TTv2 can largely correct for it until it becomes too large. On the other hand, AGAD is not affected by the offsets at all and typically shows best performance (Fig. 5B–D).

We found that even without offsets, both algorithms outperform the state-of-the-art TTv2. However, this is largely due to the choice of parameter settings which has larger writing rates onto the $\check{A}$ matrix ($l_{\max} = 5$). When using reduced rates ($l_{\max} = 1$) for devices with smaller number of states, all algorithms are fairly similar (see Supplementary Fig. 7A).

We further find that the gradients are computed so well for the proposed algorithms in spite of the offsets and transients on $\check{A}$, that the second-order effect of not correcting for the SP of $\check{W}$ (as illustrated in Fig. 2) is becoming prevalent. Indeed, the test error improves beyond the FP test error for both c-TTv2 and AGAD when the SP of $\check{W}$ is subtracted and thus corrected for (Fig. 5 closed symbols), but

increases somewhat if not (open symbols). AGAD shows better performance over c-TTv2 for larger number of states (Fig. 5A).

Although these three benchmark networks have been used extensively in previous studies on AIMC training algorithm evaluation, they are relatively small in terms of free parameters (235 K, 80 K, and 77 K, respectively). Simulating every update pulse for each weight element accurately in larger networks remains challenging due to simulation time limitations, in particular when multiple training runs are necessary for hyper-parameter tuning. However, to confirm whether the general trend of the effect of a reference value offset on the various algorithms is preserved in larger DNNs, we conducted a brief training experiment on a vision transformer[36] for classifying the CIFAR10[37] image data set, which is significantly larger (4.3M parameter; see Supplementary Methods Sec. C.1.4 for details). Indeed, even without hyper-parameter tuning, we found that when the reference offset is not perfectly corrected for, the classification error remains markedly stable only for the proposed algorithmic improvements c-TTv2 and AGAD but not for TTv2 (see Supplementary Methods Sec. C.1.4 and Supplementary Fig. 6). This is very consistent with the observed trend for the smaller benchmark DNNs (compare to Fig. 5B–D).

**Device material requirements**

The proposed AIMC training algorithms are in principle agnostic to the choice of the device material, as long as the devices support incremental bi-directional update. However, each algorithm has certain requirements on device behavior to successfully converge in the DNN training. The baseline TTv2 as well as the proposed c-TTv2 algorithm

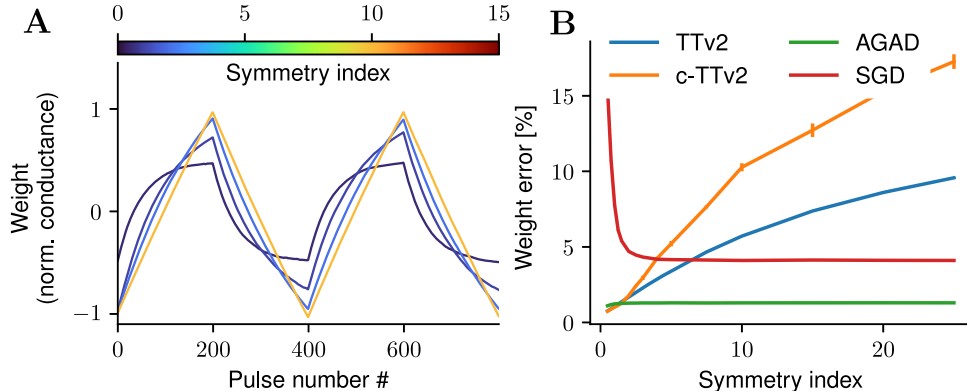

**Fig. 6 | Varying device asymmetry.** Different device materials show different degrees of asymmetric conductance responses. **A** different device responds with varying degrees of asymmetry (changing $w_{max}$ and fixing the step size). Colors of the example pulse responses to 200 up and 200 down pulses indicate the asymmetry device setting. **B** Weight errors (computed as in Fig. 4) achieved by the various algorithms depend on the degree of device symmetry. Note that only AGAD retains a very low error independent of the asymmetry setting (green line).

Asymmetry, typically very detrimental for direct SGD implementation (red line), is necessary for TTv2 (blue line) as well as c-TTv2 (orange line). This is because the latter algorithms hinge on the assumption that the conductance quickly returns to the symmetry point (SP) and the time constant to reach the symmetry point (SP) for random updates depends on the asymmetry (see Eq. (7)). Error bars indicate standard errors over 3 construction seeds.

indeed require asymmetric conductance response that is induced by the gradual saturation of the update magnitude when approaching the bounds at least for the $\check{A}$ devices (ie. the assumption of the soft-bounds model Eq. (4) must be valid). This becomes evident when repeating the same weight programming task of Fig. 4 but now varying the asymmetry of the devices (see Fig. 6). The asymmetry is changed by increasing the saturation bounds, but keeping the average update size $\delta$ constant at the SP, which effectively increases the number of states (see Eq. (6)) and causes a more symmetric (linear) pulse response around the SP (see example responses in Fig. 6A, e.g., blue curve versus orange curve, where the latter has high symmetry in up and down direction around zero). Note that the weight programming error sharply improves with higher symmetry for in-memory SGD (see Fig. 6B, red curve), however, the weight error decreases significantly for higher symmetry for TTv2 and c-TTv2 (blue and orange curves, respectively) showing that a certain amount of device asymmetry is necessary for these algorithms. In contrast, the achieved weight error of AGAD does not depend on the device asymmetry setting (Fig. 6; green line), due to its dynamic reference computation. Thus, AGAD is more widely applicable, supporting both asymmetric material choices (such as ReRAM) as well as more symmetric devices, such as capacitors or ECRAM.

**Endurance.** Another important feature of some NVM device materials (especially ReRAM materials) is the often limited endurance: after sending a very large number of voltage pulses the conductance response diminishes or fails altogether[38]. Since we propose to accumulate the gradient using fast in-memory compute, high endurance is critical. Indeed, if one counts the maximal number of pulses (positive and negative) for any of the devices used for training a DNN up to convergence (here LeNeT on MNIST; see Fig. 5) one finds values between 0.5 to 4 pulses maximally per input sample for the $\check{A}$ devices (depending on the device and hyper-parameter settings for AGAD). However, different analog crossbar arrays $\check{A}$, $\check{R}$, and $\check{W}$ (see Fig. 1 and Supplementary Fig. 2) serve very different functions and thus have very different endurance requirements. For instance, if one counts the number of pulses written onto $\check{W}$ for the same DNN training simulations, one instead finds values between $2\cdot10^{-4}$ and $4\cdot10^{-4}$ pulses maximally per input sample. Thus, the devices representing the weight $\check{W}$ require 4 orders of magnitudes less number of pulses than those used for the gradient accumulation $\check{A}$. Given that a typical training data set can have millions of examples and a fair number of epochs are

typically trained, the endurance of $\check{A}$ needs to be very high, whereas endurance requirements for the device material used for $\check{W}$ and $\check{R}$ are much less concerning.

**Retention.** Similarly, the retention requirements are vastly different for $\check{A}$, $\check{R}$ and $\check{W}$. We here define retention as the time the conductance level stays nearby the target level without external inputs. For the reference device $\check{R}$, the retention requirements can be assessed by the tolerable reference value offset. As seen from the simulations in Fig. 4B, if the reference value would drift by more than 5% from the programmed value (in percent of the conductance range, corresponding to $\sigma_r = 0.1$), during the time of the training, the TTv2 algorithm will not converge to the desired accuracy. However, for c-TTv2 the retention requirement on $\check{R}$ is significantly relaxed as the $\check{R}$ could drift up to 25% ($\sigma_r = 0.5$; Fig. 4C) within the time needed for training. However, in practice retention should be much higher, since the writing of $\check{R}$ would need to be refreshed for the next DNN training leading to inefficiencies. Since AGAD is independent of any offsets on $\check{R}$ (Fig. 4D), the programmable reference device is not needed as discussed above.

The retention requirement for $\check{W}$, on the other hand, is similar for all algorithms and on the order of the duration for a full training run, as these devices represent the converged DNN weights.

Interestingly, the retention required for $\check{A}$ is significantly less than the duration of the training. As shown in Supplementary Fig. 8, the required retention duration for $\check{A}$ in AGAD is on the order of the transfer period $Nn_s$, where $N\times N$ is the assumed matrix size, which in typical cases corresponds to the time duration the learning algorithm takes to process on the order of 100 to 1000 input samples. Since the number of training examples is often on the order of many millions, the retention requirement of $\check{A}$ is orders of magnitudes smaller than the time it takes to train the DNN. However, because of chip design considerations $\check{R}$ and $\check{A}$ likely need to be made of the same material and the retention requirements for $\check{R}$ is considerably higher. Therefore, the benefit of reduced retention for $\check{A}$ can only be exploited for AGAD, which does not need a programmable reference $\check{R}$. In this case, $\check{A}$ could be made of a high endurance but low retention material (or using an appropriate capacitor).

## Performance
In the following, we estimate the expected runtime performance for the different algorithms as well as needed memory and bandwidth.

**Table 1 | Complexity and estimated runtime performance of the weight update during DNN training**

| Algorithm | TTv2 | c-TTv2 | AGAD | in-memory SGD | MP |
|---|---|---|---|---|---|
| Storage [byte] | | $\mathcal{O}(N^2 + 2N)$ | $\mathcal{O}(3N^2 + 2N)$ | $\mathcal{O}(2N)$ | $\mathcal{O}(N^2 + 2NB)$ |
| Input loads [bit] | | $\mathcal{O}(16N + 2Nl_{\max})$ | | | $\mathcal{O}(16N)$ |
| Memory ops [bit] | $\mathcal{O}(16N/n_s)$ | $\mathcal{O}(18N/n_s)$ | $\mathcal{O}(50N/n_s)$ | $\mathcal{O}(1)$ | $\mathcal{O}(16N^2/B)$ |
| FP ops. | | $\mathcal{O}(2N + 2N/n_s)$ | $\mathcal{O}(2N + 6N/n_s)$ | $\mathcal{O}(2N)$ | $\mathcal{O}(2N^2 + N)$ |
| Analog ops [time] | | $(l_{\text{avg}} + 1/n_s)\,t_{\text{single-pulse}} + t_{\text{MVM}}/n_s$ | | $l_{\text{avg}}\,t_{\text{single-pulse}}$ | $N/B\,t_{\text{single-pulse}}$ |
| $\approx\sum$ Time est. [ns] | 56.3 | 56.3 | 62.1 | 30.9 | 3024.5 |

Complexity and average runtime of the weight update with input vector **x** and gradient input vector **d** of size $N$ (and assuming part of a mini-batch of size $B$). Number comparisons, resetting, sign flips, and ceilings are not counted. Here only FP multiplication and additions are counted as operations. The integer $n_s \geq 1$ is the period of transfer. Note that $\mathcal{O}(1)$ load and store operations (e.g., counters and constants) are omitted. An 8 bit FP number format is assumed for all digital numbers. For the time estimates, we assume $N = 512$, and FP operations compute time is calculated assuming throughput of 0.7 TFLOPS (70% utilization of 1 core of[40] (1 GFLOPS, FP8) shared for 4 analog crossbars[39]) and $n_s = 2$. Memory access time estimates are assumed to be included in the throughput and hidden with compute[40]. For the analog compute time, we assume $l_{\text{avg}} = 5$, $t_{\text{MVM}} = 40$ ns[39] and $t_{\text{single-pulse}} = 5$ ns. We further assume $B = 100$ for the total time estimate of MP. Stochastic number generation (possibly in hardware circuitry) is assumed to be parallel and hidden among the other operations.

We focus on evaluating how much time the update pass (including gradient accumulation) would take on average per input sample, since other phases, namely forward and backward phases, are identically shared among all algorithms discussed here. Note that by focusing on the update performance per input sample, we assume that the convergence behavior for the different algorithms is not vastly different in respect to the FP baseline. In other words, we assume that a similar amount of training epochs are needed to reach acceptable accuracy. We confirmed that the number of epochs needed for convergence is indeed on the same order of magnitude in respect to the FP baseline in practice (see Supplementary Fig. 5 for example traces for the data in Fig. 5A), validating our assumption in first-order approximation.

Table 1 lists the detailed runtime estimates and complexities for the proposed algorithms (see "Methods" section for detailed derivations). As additional comparison, we have listed the Mixed-Precision (MP) algorithm[20], where the gradient accumulation is done in digitally using a FP matrix. When an element of this gradient accumulation matrix reaches a threshold, pulses are sent to the (full) analog weight matrix $\breve{W}$. Thus, the number of FP operations is on the order of $\mathcal{O}(2N^2 + N)$, as one multiplication and one addition is needed per matrix element and input sample and additionally one of the input vectors needs to be scaled with the learning rate. We assume for MP that writing the full analog weight matrix is only done once per batch $B$, so that the analog time needed per input sample is $N/B\,t_{\text{single-pulse}}$ for programming $N$ rows.

As a second baseline, we compare to in-memory SGD (as described in "Methods" section "In-memory outer-product update"), which, however, yielded poor accuracy results in Fig. 5.

When one assumes that a certain amount $X$ of digital compute throughput is available exclusively for a single analog crossbar array, then we can estimate the average time (per input sample) the gradient update step would take. For approximate numbers, we assume that a single update pulse would take approximately 5 ns, a single MVM about 40 ns[39], and that the memory operations (Table 1, rows in first section) can be hidden behind the compute[40]. In Supplementary Fig. 9, the average time for an update is plotted against the amount of available compute. As seen from Table 1, if one assumes a state-of-the-art number of 175 billion FP operations per second (FLOPS) (that is 0.7 TFLOPS[40], shared among 4 crossbar arrays), the proposed algorithms out-perform the alternative MP algorithm by a large margin, showing the benefits of AIMC for in-memory gradient update (about 50× faster, even if one already assumes a batch size of 100, which favors the MP algorithm). Moreover, computing the gradient in digital requires a much higher memory throughput for MP (see row "Memory ops" in Table 1), which could be challenging to maintain. Since at most one row (or column) is processed in digitally for our proposed algorithms per input, memory bandwidth is not a bottleneck.

Note that for these numbers we have considered a conservative setting of the hyper-parameters, $n_s = 2$ and $l_{\max} = 5$. In fact, the runtime of the algorithms TTv2, c-TTv2, and AGAD would all converge to the limit of in-memory SGD with increasing values of $n_s$, as their additional compute all scale with $\frac{1}{n_s}$ (see Table 1 "FP ops" and "Analog ops"). We find that higher $n_s$ numbers are supported, however, accuracy drops slightly if $n_s$ gets too high (depending on the matrix size), if at the same time, the device number of states is limited (see Supplementary Fig. 7 for the effect of different $n_s$ settings during DNN training). Note that if $n_s$ increases, the analog devices $\breve{A}$ have to accumulate and hold the information for more input samples before being read out. However, as shown in Supplementary Fig. 7A, DNNs can also be trained with e.g., $n_s = 10$ and $l_{\max} = 1$ without accuracy loss with certain device characteristics (here $n_{\text{states}} = 20$). With the same digital throughput assumptions as above, the expected update time for AGAD in Table 1 would then further reduce to 17.1 ns reaching an acceleration factor of about 175× compared to MP (see Table 1; see also Supplementary Fig. 9 for more parameter settings).

Finally, as detailed in the "Methods" section "AGAD algorithm", one could also set $\beta = 1$ in AGAD which would make the storing and computing of $P$ unnecessary, saving $\mathcal{O}(N^2)$ storage and $\mathcal{O}(3N/n_s)$ compute for the estimation of the leaky average. However, we find that accuracy is generally improved when setting $\beta < 1$ depending on the number of available states $n_{\text{states}}$ (see Supplementary Fig. 7, red line labeled AGAD with $\beta = 1$).

## Discussion

We have introduced two learning algorithms for fast parallel in-memory training using crossbar arrays. In this approach, the weight update necessary for the stochastic gradient descent is directly done in-memory using parallelly pulsed increments for adding the outer product between the activations and backpropagated error signals to the weights.

Note that this in-memory training approach is radically different from hardware-aware training typically employed when using analog crossbar arrays for DNN inference only (e.g.,[32,41,42]). In the latter case, the DNN weights are (re)-trained in software (using traditional digital CPUs or GPUs) assuming generic noise sources to improve the noise robustness. The final weights are programmed once onto the analog AI hardware accelerator which is then used in an inference application without further training. In contrast, in our study the training of the weights itself is done by the analog AI hardware accelerator in-memory on the crossbar arrays, thus opening up the possibility for high energy efficiency during the training of DNNs. Whether inference is then done with the same hardware using the trained weights depends on the application. While directly using the trained weights with the same hardware for inference would be the most efficient, other approaches are possible as well. For instance, Gokmen[24] suggests extracting the

trained weights during in-memory training using stochastic weight averaging in a highly efficient way, so that they can then be used for any other hardware during inference, including reduced precision digital inference accelerators. Other analog inference hardware could be used as well, however, an additional programming error penalty will be introduced in this case. Nevertheless, given that realistic device noise is naturally present during our proposed in-memory training, the resulting weights are likely to be robust to any device noise in a way similar to the conventional hardware-aware training approach in software (see e.g., ref. [32]).

For our algorithms, we found that the converged accuracy matches or exceeds the current state-of-the-art in-memory training algorithm TTv2[24]. Indeed, in cases where the TTv2 algorithm suffers severe convergence issues, the proposed algorithms are considerably improved. In particular, TTv2 suffers if the reference conductance is not programmed very precisely (within few percent of the conductance range), which hasn't been considered during its conception[24]. A precise writing of the reference is very difficult to achieve with current device materials rendering the application of TTv2 unrealistic, in particular for larger-scale DNN training. Both proposed algorithms, c-TTv2, and AGAD, relax this requirement significantly.

The computational complexity added to TTv2 for the proposed algorithms is negligible for c-TTv2. While AGAD introduces slightly more digital compute and storage, the overall runtime is nevertheless expected to be still orders of magnitude faster than alternatives, where the gradient matrix is computed in digital and therefore scales with $\mathcal{O}(N^2)$[20]. Indeed, when estimating the average gradient update time for a $512 \times 512$ weight matrix in Table 1 with reasonable assumptions, we find 62.1ns for AGAD versus >3000 ns when updating the gradient matrix in digital instead. This large improvement is achieved because the in-memory update pass uses only linear order of digital operations ($\mathcal{O}(N)$) with the proposed algorithms. Moreover, since the weight is stored in analog memory, the forward and backward passes can be accelerated as well. While the MVMs needed for the forward and backward passes can be accelerated in-memory in constant time ($\mathcal{O}(1)$), there are, however, typically other utility $\mathcal{O}(N)$ computations done in digital besides the mere MVMs. For instance, rescaling of the input and outputs for improving the AIMC MVM fidelity (see e.g., ref. 32 for a discussion), or computing other layers such as affine transforms of normalization layers, skip connections, activation functions, that are all part of modern DNNs. Since these utility layers commonly have at least $\mathcal{O}(N)$ runtime complexity, the additional $\mathcal{O}(N)$ digital operations needed for the proposed updated passes will not change the overall runtime complexity of the full training, which includes forward, backward, and update passes[24].

We like to emphasize that the reduced number of digital operations necessary for our AIMC training algorithm, together with the non-von-Neumann architecture and high energy efficiency of MVMs and outer products on the analog crossbar arrays, translates into a highly energy-efficient approach for DNN training in comparison to traditional digital ways of compute. While the energy efficiency per digital operation has improved over time[43], the complexity of the memory access and MVM compute still remains bounded by $\mathcal{O}(N^2)$ and is thus inherently worse than our AIMC approach. Indeed, even more energy savings could result from co-designing DNNs for deployment on AIMC architectures, as the scaling laws of the SGD training is different compared to digital hardware. For instance, a large and dense matrix multiplication is much less costly on AIMC than on digital von Neumann hardware, potentially opening up opportunities for designing novel energy-efficient DNN architectures with high accuracy tailored to AIMC in the spirit of[44].

We here have given a runtime estimate for the gradient update only instead of a complete estimate of the time needed to train a DNN on a given chip. A complete estimate has to take into account many details of the mixed analog-digital chip architecture, as it needs to consider not only the forward pass computations of all analog and digital auxiliary layers (as recently shown for an energy estimate for inference-only AIMC hardware[39]), but also the backward pass, and weight update computations that require intermediately storing of results (see ref. 24 for a discussion). Therefore, a complete energy estimate for a full DNN training run has to be based on a specific AIMC chip architecture and is thus beyond the current study.

The hallmark of AGAD is to compute the reference value on-the-fly. Interestingly, even in the field of analog amplifier design, it has been previously proposed to dynamically compute the zero point (auto-zero) in conjunction with the chopping technique. This combination as been shown to have superior performance in challenging signal-processing application[45]. This approach is qualitatively similar to AGAD that employs both the chopper as well as an on-the-fly reference.

Note that the reference value computed for AGAD is different from the reference value programmed onto the conductances $\check{R}$ in case of TTv2 and c-TTv2. In the latter case, the symmetry point (SP) of $\check{A}$ is used as reference together with a differential read of both conductances. Consequently, TTv2 and c-TTv2 make in practice quite restrictive assumptions on the device model, namely that an unique SP exists, which is moreover stable over time. In contrast, AGAD subtracts an estimate of the history of the transient conductance value that was reached before the chopper sign flipped in digital. This digitally stored reference value is based on the transient conductance dynamics and thus independent from any SP assumption. Using this transient on-the-fly reference value computation is made possible by the introduction of the chopper that changes the sign and thus the direction of the information accumulation. Given that the devices have limited conductance range, incoming gradients therefore can use the full dynamics range effectively.

The on-the-fly reference value computation has several advantages for AIMC DNN training. First, the lengthy estimation and programming of the reference arrays $\check{R}$ prior to the DNN training run is not necessary thus simplifying and improving the overall training process. Second, the chip design is simplified as the differential read of two devices does not needed to be implemented in circuitry. Third, the unit cell of the crossbar array is simplified because no individual and programmable reference for each element in the weight matrix is needed at all, saving considerably in hardware complexity and chip area cost.

Finally, the AGAD algorithm greatly broadens the device material choices. The on-the-fly reference estimation enables the computation on transients, meaning that the average conductance level becomes irrelevant. This means that both symmetric or asymmetric devices can be used similarly well for the gradient accumulation. This contrasts with TTv2 and c-TTv2, which are designed specifically for and require asymmetric device conductance responses. Enabling such broad device material choice is important for future applicability of AIMC for DNN training. For instance, very high endurance ReRAM (many millions of pulses) is beyond the current state-of-the-art for this material choice, however, other material choices exist such as ECRAM, or capacitors, that essentially have no endurance limit, but have a much more symmetrical response. We also show that gradient accumulation material only needs to show very short retention, thus further relaxing the material requirements of AGAD. In conclusion, we show that both c-TTv2 and AGAD push the boundary of in-memory training performance, while considerably relaxing device material and chip design requirements, opening a realistic path for accelerating DNN training using analog in-memory computation.

## Method

### Analog matrix-vector multiplication

Using resistive crossbar arrays to compute an MVM in-memory has been suggested early on[46], and multiple prototype chips where MVMs

of DNNs during inference are accelerated have been recently described[6–9,11,47]. In these studies, the weights of a linear layer are stored in a crossbar array of tunable conductances, inputs are encoded e.g., in voltage pulses, and Ohm's and Kirchhoff's laws are used to multiply the weights with the inputs and accumulate the products (Supplementary Fig. 1A, see also e.g., ref. 1 for more details). In many designs, the resulting currents or charges are converted back to digital by highly parallel analog-to-digital converters (ADCs).

For fully in-memory analog training, as suggested in ref. 13, additionally a transposed MVM has to be implemented for the backward pass, which can be achieved by transposing inputs and outputs accordingly (see Supplementary Fig. 1B).

Here, we simulate the non-linearity induced by an MVM in the forward and backward following previous studies[13]. We use the standard forward and backward settings in the simulation package (AIHWKIT)[29], which includes output noise, input, and output quantization, as well as bound and noise management techniques as described in[33] (see Supplementary Methods Sec. C.1 for the exact AIMC MVM model settings).

However, we focus on the nonidealities induced by the incremental update of the conductances (as detailed below) which are typically much more challenging for AIMC training than the MVM nonlinearities. For instance, it has recently been shown in simulation that with realistic MVM assumptions many large-scale DNNs can be deployed without significant accuracy drop on AIMC inference hardware when retrained properly[32].

### In-memory outer-product update

While accelerating the forward and the backward pass of SGD using AIMC is promising, for a full in-memory training solution, in-memory gradient computation and weight update have to be considered for acceleration as well.

For the gradient accumulation of and $N \times N$ weight matrix $W$ of a linear layer (i.e., computing $\mathbf{y} = W\mathbf{x}$), the outer-product update $W \leftarrow W + \lambda \, \mathbf{d}\mathbf{x}^T$ needs to be computed. While this can be done in digital, possibly exploiting sparseness (e.g., MP, see ref. 20), it would still require on the order of $\mathcal{O}(N^2)$ digital operations, and doing so would thus limit the overall acceleration factor obtainable for in-memory training. To accelerate also the outer-product update to be performed in-memory and fully parallel, Gokmen & Vlasov[13] suggested to use stochastic pulse trains and their coincidence (as illustrated in Supplementary Fig. 1C).

The exact update algorithm has gone through a number of improvements in recent years[33,35], however, we here use a yet improved version in Supplementary Alg. 1. In particular, we suggest to dynamically adjust the pulse trains in length for better efficiency. Note that we assume in Supplementary Alg. 1 for simplicity of the formulation that a mixture of negative or positive pulses across inputs $x_i$ are possible, while in practice, negative and positive pulses are sent sequentially in two separate phases (setting all $x_i < 0$ to 0 in the first phase and all $x > 0$ to zero in the second).

In the "Results" section, we will compare the performance of our in-memory training algorithms in more detail, and they are partly based on this outer product. Note that the Supplementary Alg. 1 takes $\mathcal{O}(2N)$ FP operations for each vector update (assuming a vector length of $N$) to compute the absolute maximal values that is needed to scale the probabilities. Then maximally $l_{max}$ pulses are given (in each of the two sequential phases of negative and positive pulses), however, the dynamical adjustment of the pulse train length (see Supplementary Alg. 1) leads to only $l_{avg} \leq l_{max}$ pulses on average over input vectors. Thus, assuming a pulse duration of $t_{single-pulse}$, the runtime complexity of digital compute of the output product update is $\mathcal{O}(2N)$ and the average time for the analog part is $2t_{single-pulse}l_{avg}$. For the pulsing, $2Nl_{avg}$ stochastic numbers are generated or $2Nl_{max}$ pre-generated pseudo-random pulse trains are loaded from memory, and therefore a complexity of the memory loads is $\mathcal{O}(2Nl_{max})$ bits. If one further

assumes that the input and output vectors, $\mathbf{x}$ and $\mathbf{d}$ need to by transiently stored to compute the pulse probabilities (e.g., in 8-bit FP format), then the overall memory operations required is on the order of $\mathcal{O}(2Nl_{max} + 16N)$ bits.

Previous studies[13,33,35] have investigated the noise properties when using Supplementary Alg. 1 to directly implement the gradient update in-memory and it turns out that this would require very symmetric switching characteristics of the memory device elements in particular for large DNNs[22]. Thus, the requirements of such an in-memory SGD algorithm turns out to be too challenging in face of the asymmetry observed in today's device materials, which we discuss in the next section.

### Device material model

When subject to a large enough voltage pulse, bi-directionally switching device materials, such as ReRAM[15], ECRAM[16,17], or capacitors[18], show incremental conductance changes. In previous studies[48,49], it was shown that the soft-bounds model characterizes the switching behavior of such materials qualitatively well. According to that model, the conductance change $g \leftarrow g + \Delta g_D$ to a single voltage pulse in either up ($D = +$ or down $D = -$) direction is given by

$$\begin{aligned} \Delta g_+ &\equiv \alpha_+ \left( g_{max} - g \right) \\ \Delta g_- &\equiv \alpha_- \left( g_{min} - g \right) \end{aligned} \tag{2}$$

where thus the induced conductance change gradually reduces towards the conductance bounds. While here the conductance is measured in physical units, it is more convenient for the following discussion to (arbitrarily) normalize the conductances. For that, we first set $g_{half-range} \equiv \frac{\langle g_{max} \rangle - \langle g_{min} \rangle}{2}$ where the average is taken over the individual devices (that in general have individual $g_{min}$ and $g_{max}$ values due to device-to-device variations). Then, we set the normalized conductance value to $\breve{w} \equiv \frac{g - \langle g_{min} \rangle}{g_{half-range}} - 1$, so that for a device at $\langle g_{min} \rangle$ the normalized value is $\breve{w} = -1$, and $\langle g_{max} \rangle$ corresponds to $\breve{w} = 1$, and finally $\frac{\langle g_{min} \rangle + \langle g_{max} \rangle}{2}$ corresponds to $\breve{w} = 0$.

Using this normalization, Eq. (2) becomes (assuming no device variations for the moment, i.e., $g_{max} = \langle g_{max} \rangle$ and $g_{min} = \langle g_{min} \rangle$)

$$\begin{aligned} \Delta \breve{w}_+ &\equiv \alpha_+ \left( 1 - \breve{w} \right) \\ \Delta \breve{w}_- &\equiv -\alpha_- \left( 1 + \breve{w} \right) \end{aligned} \tag{3}$$

which corresponds to the soft-bounds model in[48] albeit with a different conductance normalization (here shifted to the range of $-1, ..., 1$ instead of $0, ..., 1$ to ease of discussion of the algorithmic zero point).

We introduce device-to-device variations on the saturation levels as well as on the slope parameter $\alpha$ and cycle-to-cycle update fluctuations to arrive at the full model

$$\begin{aligned} \Delta \breve{w}_+ (\breve{w} \mid \boldsymbol{\theta}) &\equiv \alpha_+ \left( \frac{\breve{w}_{max} - \breve{w}}{\breve{w}_{max}} + \sigma_{c-to-c} \xi \right) \\ \Delta \breve{w}_- (\breve{w} \mid \boldsymbol{\theta}) &\equiv -\alpha_- \left( \frac{\breve{w}_{min} - \breve{w}}{\breve{w}_{min}} + \sigma_{c-to-c} \xi \right) \end{aligned} \tag{4}$$

where $\xi$ are standard normal random numbers (drawn for each update) to model the update fluctuations of strength $\sigma_{c-to-c}$. Here we chose to normalize the difference of the actual conductance to the bound by the bound, that is e.g., $\frac{\breve{w}_{max} - \breve{w}}{\breve{w}_{max}}$, so that the update size remains constant for the same relative distance of $\breve{w}$ towards the bounds when varying solely $\breve{w}_{min}$ or $\breve{w}_{max}$. Note that in simulations the normalized conductance values are ensured to be clamped to the saturation levels (between $\breve{w}_{min}$ and $\breve{w}_{max}$) to avoid that the additive noise would drive the conductance to not supported levels.

In Eq. (4), we use the placeholder $\boldsymbol{\theta}$ for the hyper-parameters as defined in the following. To capture device-to-device variability, we draw random variations during construction according to $\breve{w}_{max} = \max(1 + \sigma_b \xi_1, 0)$ and $\breve{w}_{min} = \min(-1 + \sigma_b \xi_2, 0)$ where $\xi_i \in$

$\mathcal{N}(0,1)$ are random numbers that are different for each device but fixed during training.

The slope parameters are given by

$$\begin{aligned} \alpha_+ &\equiv \delta\,(\gamma + \rho) \\ \alpha_- &\equiv \delta\,(\gamma - \rho) \end{aligned} \tag{5}$$

where $\gamma = e^{\sigma_{d\text{-}to\text{-}d}\xi_3}$, and $\rho = \sigma_{\pm}\xi_4$, so that $\sigma_{d\text{-}to\text{-}d}$ is a hyper-parameter for the variation of the slope across devices, and $\sigma_{\pm}$ a separate device-to-device variation in the difference of the slope between up and down direction. The material parameter $\delta$ determines the average update response for one pulse when the weight is at $\breve{w} = 0$. We define the number of device states (for a given fixed setting of the incremental update noise level $\sigma_{c\text{-}to\text{-}c}$) by the average weight range divided by $\delta$, that is

$$n_{\text{states}} = \frac{\breve{w}_{\max} - \breve{w}_{\min}}{\delta}. \tag{6}$$

We found in previous studies that this model of the device-to-device variations fits ReRAM (array) measurements reasonable well[25,50].

**Symmetry point.** It can easily be seen that for the device model Eq. (4), the conductance change in response to a positive voltage pulse linearly depends on the current conductance value and decreases up to the bound $\breve{w}_{\max}$ where it becomes zeros. Likewise, the conductance change linearly decreases for negative updates down to the bound $\breve{w}_{\min}$, and thus the (normalized) conductance $\breve{w}$ will saturate at $\breve{w}_{\max}$ and $\breve{w}_{\min}$. Because of this gradual saturating (soft-bounds) behavior, there exists a conductance value at which the up and down conductance change magnitudes are equal on average, which is called the symmetry point (SP)[23,30] and denoted as $\breve{w}^*$.

If one assumes that random up-down pulsing (without a bias in either direction) is applied to the devices, the device will reach its SP quickly. This can be easily seen in the case where a positive pulse always follows a negative pulse. Then the weight change can be written as (assuming for the moment $\sigma_b = \sigma_{c\text{-}to\text{-}c} = \sigma_{\pm} = \sigma_{d\text{-}to\text{-}d} = 0$):

$$\begin{aligned} \Delta\breve{w} &\approx \Delta\breve{w}_+(\breve{w}\,|\,\boldsymbol{\theta}) + \Delta\breve{w}_-(\breve{w}\,|\,\boldsymbol{\theta}) \\ &= -2\delta\breve{w} \end{aligned} \tag{7}$$

which shows that for repeated pairs of up-down pulses the weight will decay exponentially with (approximate) decay rate of $\tau = 2\delta$ to a fixed point at $\breve{w}^* = 0$.

Solving Eq. (4) for the SP $\breve{w}^*$ by setting $\Delta\breve{w}_-\!\left(\breve{w}^*\,|\,\boldsymbol{\theta}\right) = \Delta\breve{w}_+\!\left(\breve{w}^*\,|\,\boldsymbol{\theta}\right)$, one finds for the non-degenerated case, i.e., $\breve{w}_{\max} > \breve{w}_{\min}$, $\alpha_+ > 0$, and $\alpha_- > 0$,

$$\breve{w}^* = \frac{\alpha_+ - \alpha_-}{\frac{\alpha_+}{\breve{w}_{\max}} - \frac{\alpha_-}{\breve{w}_{\min}}} = \frac{2\rho}{\frac{\gamma+\rho}{\breve{w}_{\max}} - \frac{\gamma-\rho}{\breve{w}_{\min}}} \tag{8}$$

Note that some of the AIMC training algorithms discussed in the following will use this SP as a reference value of the gradient accumulation.

**Recap of the Tiki-Taka (version 2) algorithm**
In the TTv2 learning algorithm (see Fig. 1 for an illustration), three tunable conductance elements for each weight matrix element are required, namely the matrices $\breve{A}$, $\breve{R}$, and $\breve{W}$, where we write $\breve{X}$ for a weight matrix $X$ that is thought of coded into the conductances of a crossbar array, to distinguish between matrices that are in digital memory. The first two conductances, $\breve{A}$ and $\breve{R}$, are used to accumulate the gradient accumulation and storing the SP of $\breve{A}$, respectively, and are read intermittently in fast differential manner $\breve{A} - \breve{R}$, whereas $\breve{W}$ is used as the representation of the weight $W$ of a linear layer and thus

used in the forward and backward passes. On a functional level, the algorithm is similar to modern SGD methods that introduce a momentum term (such as ADAM[51]), since also here the gradient is first computed and accumulated in a leaky fashion onto a separate matrix before being added to the weight matrix. However, the analog-friendly TTv2 algorithm computes and transfers the accumulated gradients asynchronously for each row (or column) to gain run-time advantages. Furthermore, crucially, the device asymmetry of the memory element causes an input-dependent decay of the recently accumulated gradients as opposed to the usual constant decay rate of the momentum term that is difficult to efficiently implement in-memory (see also discussion in refs. 24, 30).

While this TTv2 algorithm greatly improves the material specifications by introducing low pass filtering of the recent gradients, it hinges on the assumption that the device has a pre-defined and stable SP within its conductance range[30]. The SP is defined as the conductance value, where a positive and a negative update will result on average in the same net change of the conductance. Because of the assumed device asymmetry, the SP acts as a stable fix point for random inputs, which causes the accumulated gradient on $\breve{A}$ to automatically decay near convergence (see "Methods" section "Symmetry point"). However, to induce a decay towards zero algorithmically, it is essential to identify the SP with the zero value for each device, which is achieved by removing the offset using a reference array $\breve{R}$ (as illustrated in Fig. 2). The reference conductance $\breve{R}$ is thus used to store the SP values of its corresponding devices of $\breve{A}$ and instead of directly reading $\breve{A}$, the difference $\breve{A} - \breve{R}$ is read, while only $\breve{A}$ is updated during training.

Taken together, for TTv2 the reference array $\breve{R}$ must be set to the SP of a corresponding analog matrix $\breve{A}$ prior to the DNN training. The algorithm of how to program $\breve{R}$ to the SP in practice is discussed in ref. 26. It turns out, however, that the programming as well as the SP estimation is in general subject to errors. To model this error, we set (with Eq. (8)) the elements of $\breve{R}$ to

$$\breve{r}_{ij} = \breve{w}^*_{ij} + \xi_{ij} \tag{9}$$

where $\xi_{ij} \in \mathcal{N}(\mu_R, \sigma_R)$. Thus $\xi_{ij}$ models the remaining error on the reference device after SP subtraction.

In more mathematical detail, to lower the device requirements for in-memory SGD, TTv2 computes the outer product update in a fast manner in-memory and thus accumulate the recent past of the gradients $(\mathbf{dx}^T)$ onto a separate analog crossbar array $\breve{A}$, but slowly transfer the recent accumulated gradients by sequential vector reads of $\breve{A}$ onto the analog weight matrix $\breve{W}$, to counter-act the loss of the information due to the device asymmetry on the gradient matrix $\breve{A}$. So, each vector update, the following three sequential operations are in principle done (see illustration in Fig. 1):

$$\begin{aligned} &\xrightarrow[\text{parallel update Supplementary Alg. 1}]{\propto\,\lambda_A\,\mathbf{dx}^T} \breve{A} \\ &\xrightarrow[\text{every } n_s \text{ an MVM read}]{\lambda_H(\breve{A}-\breve{R})\,\mathbf{v}_k} \mathbf{h}_k \xrightarrow[\text{write single pulses}]{\propto\,\lfloor\mathbf{h}_k\rfloor_0\,\mathbf{v}_k^T} \breve{W} \end{aligned} \tag{10}$$

The outer-product update onto the analog array $\breve{A}$ is done using the stochastic pulse trains and coincidences as described in Supplementary Alg. 1 and is thus essentially $\mathcal{O}(1)$. For the second step, a row of $\breve{A}$ can be read by computing an MVM in-memory by using the corresponding one-hot unit vectors $\mathbf{v}_k$ as input and is thus fast ($\mathcal{O}(1)$). Note that instead of reading a row as described, one could similarly read out a column of $\breve{A}$ instead by using the transposed read capability—as is true for the other algorithms that are described below. To not confuse the description, we will here explain only the case of rows with the understanding that instead of rows columns could be processed as well.

The resulting FP vector $\mathbf{z}_k = (\breve{A} - \breve{R}) \mathbf{v}_k$ is multiplied with a learning rate $\lambda_H$ and then added onto the corresponding row of the digital FP matrix $H$. The selected row $k$ could be random, or sequentially iterated through all rows with wrapped boundaries. Each time a transfer is made, the absolute vector values $|\mathbf{h}_k|$ are tested against a threshold (typically set to 1), and single pulses are used to update the corresponding row of the analog weight matrix $\breve{W}$ when the threshold is reached. Thereby the sign of $h_{ik}$) is respected (note that we use the floor-towards-zero sign $\lfloor \mathbf{h}_k \rfloor_0$). This writing of single pulses can be done in $\mathcal{O}(1)$ as only one column is written in parallel.

This TTv2 algorithm (as described in all details in Supplementary Alg. 2 with $\rho = 0$) is our baseline comparison.

**Overall performance.** The average runtime complexity of the TTv2 algorithm per input sample is divided into digital operations (compute and storage) and time for the analog operations. As detailed in the "Methods" section "In-memory outer-product update", the outer product into $\breve{A}$ needs $\mathcal{O}(2N)$ digital operations. The additional $\mathcal{O}(N)$ scaling and $\mathcal{O}(N)$ additions needed for the transfer of the readout of $\breve{A}$ to the digital matrix $H$ are only done every $n_s$ vector updates and skipped otherwise, so that the average complexity of digital operations per input vector sample is $\mathcal{O}(2N/n_s)$. Similarly, the writing onto $\breve{W}$ is only executed every $n_s$ inputs. Altogether, the average complexity of digital operations for the full gradient update is thus $\mathcal{O}(2N(1 + \frac{1}{n_s}))$.

The average analog runtime per input sample is $2(l_{avg} + \frac{1}{n_s}) t_{single-pulse} + \frac{1}{n_s} t_{MVM}$, given that at most 2 pulses (positive and negative phase) are sent for the write on $\breve{W}$ and one read (forward pass) of $\breve{A}$ has to be performed (with time $t_{MVM}$) every $n_s$ input samples. Note that although $\mathcal{O}(8N^2)$ bit memory is needed to store $H$, only $\mathcal{O}(16N/n_s)$ bit memory operations (load and store per input sample) are needed in addition to those needed the outer product on $\breve{A}$ (see Methods section 'In-memory outer-product update'), as only one row is operated on for the transfer and writing, which could thus be prefetched and cached efficiently.

**Fast and robust in-memory training**
We propose two algorithms based on TTv2, that improve the gradient computation in case of any kinds of reference instability or residual offsets. Both algorithms introduce a technique borrowed from amplifier circuit design, called chopper[27]. A chopper is a well-known technique to remove any offsets or residuals caused by the accumulating system that are not present in the signal, by modulating the signal with a random sign-change (the chopper) that is then corrected for when reading from the accumulator.

**Chopped-TTv2 algorithm.** While using a reference matrix $\breve{R}$ has the advantage to subtract the SP from $\breve{A}$ efficiently using a differential read, the design choice comes with unique challenges. In particular, the programming of $\breve{R}$ might be inexact, or the SP might be wrongly estimated or vary on a slow time scale. As shown in the "Results" section, any residual offsets $o_r \equiv \breve{r} - \breve{a}^*$ would constantly accumulate on $H$ and be written onto $\breve{W}$ thus biasing the weight matrix unwantedly. Moreover, the decay of $\breve{A}$ to its SP is the slower the more states the device has and input dependent (see Eq. (7)). While feedback from the loss would eventually change the gradients and correct $\breve{W}$, the learning dynamics might nevertheless be impacted.

For robustness to any remaining offsets and low-frequency noise sources, we suggest here to improve the algorithm by introducing choppers. Chopper stabilization is a common method for offset correction in amplifier circuit design[27]. We use choppers to modulate the incoming signal before gradient accumulation, and subsequently demodulate during the reading of the accumulated gradient.

In more detail, we introduce choppers $c_j \in \{-1, 1\}$ that flip the sign of each of the activations $x_j$ before the gradient accumulation on $\breve{A}$, that is $c_j x_j$ (or in vector notation with element-wise product $\mathbf{c} \odot \mathbf{x}$).

When reading the $k$-th row of $\breve{A}$ to be transferred onto $H$, we apply the corresponding chopper $c_k$ to recover the correct sign of the signal. Thus, the overall structure of the update remains the same as illustrated in Fig. 1, however, it is now set $\hat{\mathbf{x}} \equiv \mathbf{c} \odot \mathbf{x}$ and $\mathbf{z}_k \equiv c_k (\breve{A} - \breve{R}) \mathbf{v}_k$.

In summary, the gradient update now becomes (compare also to Supplementary Fig. 2)

$$\xrightarrow[\text{parallel update Supplementary Alg. 1}]{\propto \lambda_A \, \mathbf{d}(\mathbf{c} \odot \mathbf{x})^T} \breve{A}$$

$$\xrightarrow[\text{every } n_s \text{ an MVM read}]{c_k \lambda_H (\breve{A} - \breve{R}) \mathbf{v}_k} \mathbf{h}_k \xrightarrow[\text{write single pulses}]{\propto \lfloor \mathbf{h}_k \rfloor_0 \mathbf{v}_k^T} \breve{W} \tag{11}$$

The choppers are flipped randomly with a probability $\rho$ every read cycle (see Supplementary Alg. 2 for the detailed algorithm). In this manner, any low frequency component that is caused by the asymmetry or any remaining offsets and transients on $\breve{A}$ is not modulated by the chopper and thus canceled out by the sign flips. We call this algorithm Chopped-TTv2 (c-TTv2) stochastic gradient descent.

**Overall performance.** Since only sign changes are introduced the c-TTv2 algorithms has largely the same runtime performance numbers as the baseline TTv2 (see "Methods" section "Recap of the Tiki-Taka (version 2) algorithm"). Since applying and flipping a sign is very fast, we omit these operations, however, the current signs must be loaded and stored every $n_s$ input samples, so that the average number of memory operations per input sample increases by $2N/n_s$ bits.

**AGAD algorithm.** While the chopper together with the low-pass filtering greatly improve the resilience to any remaining offsets (see "Results" section), if offsets become too large simply low-pass filtering will not be effective enough.

Moreover, if the training was perfectly inert to any offsets $o_r$ then the differential read could be replaced by a direct read of $\breve{A}$ (using constant reference conductance to balance the currents), which would significantly reduce the chip design complexity and the chip area needed for $\breve{R}$. In addition, the SP of $\breve{A}$ would neither need to be estimated nor programmed, improving handling in practice.

To address these issues, we suggest the recent history of the transient conductance dynamics as reference instead of the troublesome programming of predetermined values that depend on the individual device characteristics. In more detail, we propose to use choppers as in c-TTv2, so again $\hat{\mathbf{x}} \equiv \mathbf{c} \odot \mathbf{x}$ in Fig. 1, however, now we set $\mathbf{z}_k \equiv c_k (\breve{A}' \mathbf{v}_k - \mathbf{p}_k^{ref})$. where we use additional digital compute and memory to store a digital reference matrix $P^{ref}$. Note that the readout from $\breve{A}'$ could be simply a direct readout of $\breve{A}$ since $P^{ref}$ is used as reference values. The additional conductance $\breve{R}$ are thus not needed. However, to align the comparison with the other algorithms, we use $\breve{A}' \equiv \breve{A} - \breve{R}$ in the numerical simulations.

With that, the schematics becomes

$$\xrightarrow[\text{parallel update Supplementary Alg. 1}]{\propto \lambda_A \, \mathbf{d}(\mathbf{c} \odot \mathbf{x})^T} \breve{A}$$

$$\xrightarrow[\text{every } n_s \text{ an MVM read}]{c_k \lambda_H (\breve{A}' \mathbf{v}_k - \mathbf{p}_k^{ref})} \mathbf{h}_k \xrightarrow[\text{write single pulses}]{\propto \lfloor \mathbf{h}_k \rfloor_0 \mathbf{v}_k^T \rightarrow} \breve{W} \tag{12}$$

To set the digital reference matrix $P^{ref}$, another digital matrix $P$ is computed row-by-row as an leaky average of the recent past readouts of the $k$-the row of $\breve{A}'$, i.e., $\boldsymbol{\omega} \equiv \breve{A}' \mathbf{v}_k$:

$$\mathbf{p}_k \leftarrow (1 - \beta) \mathbf{p}_k - \beta \boldsymbol{\omega} \tag{13}$$

where $0 \leq \beta \leq 1$ is the time constant of the leaky average. Then the reference matrix (row) is set $\mathbf{p}_k^{ref} \leftarrow \mathbf{p}_k$ only when the chopper sign $c_k$

flips. The chopper flips could be either randomly (with probability $\rho$) or at a fixed period of readouts of row $k$.

The reasoning of Eq. (13) is that the chopper flips are unrelated to the direction of gradient information. Therefore, if a significant average gradient is currently present, the direction of updates onto $\breve{A}$ has to change its direction when the chopper flips. Thus, the recent past values of $\breve{A}$ before the sign flip can serve as good reference point for the following chopper period until the next sign flip.

This algorithm is called (AGAD). See Supplementary Alg. 3 and Supplementary Fig. 2 for implementation details. Note that here two additional FP matrices $P$ and $P^{\text{ref}}$ need to be stored in local memory. However, it is possible to reduce the requirement to one matrix $P^{\text{ref}}$ if the leaky average of the recent past Eq. (13) is omitted and only the previous readout is used instead (that is when formally $\beta = 1$ in Eq. (13)). See "Results" section for a discussion of these choices.

**Overall performance.** The AGAD algorithm only introduces additional digital compute in the transfer cycle. Thus, the runtime performance and analog compute of c-TTv2 still holds. However, to subtract the vector $\mathbf{p}^{\text{ref}}$ it needs $\mathcal{O}(N/n_s)$ additional digital operations per input sample. Moreover, if $\beta \neq 1$, then extra $\mathcal{O}(3N/n_s)$ digital operations (two scaling and one addition) are needed for updating $\mathbf{p}$. In terms of memory, the digital matrix $P^{\text{ref}}$ needs $8 \cdot N^2$ bits memory storage. Additionally, $P$ needs $8 \cdot N^2$ bits memory storage as well if used in case of $\beta \neq 1$. The number of memory operations per input sample also increases by $\mathcal{O}(8 \cdot 2N/n_s)$ or $\mathcal{O}(8 \cdot 4N/n_s)$, respectively, when $\beta = 1$ or $\beta \neq 1$, for loading and storing the additional rows $\mathbf{p}$ and $\mathbf{p}^{\text{ref}}$.

**Determining the learning rates**

In the original formulation of the TTv2 algorithm[24], the learning rate $\lambda_H$ for writing onto the hidden matrix $H$ was not specified explicitly (compare to Fig. 1). We here suggest to use

$$\lambda_H = \frac{\lambda \, n_s \, n}{\gamma_0 \delta_{\breve{W}} \lambda_A} \qquad (14)$$

where $n$ is the number of rows of the weight matrix, $n_s$ the number of gradient updates done before a single row-read of $\breve{A}$, and $\delta_{\breve{W}}$ is the average update response size at the SP of $\breve{W}$ (see "Methods" section "Device material model").

Here $\lambda$ is the learning rate of the standard SGD, which might be scheduled. Note that we thus scale $H$ by the overall SGD learning rate and not the writing onto $\breve{A}$. The hyper-parameter $\gamma_0$ specifies the length of accumulation, with larger values averaging the read gradients for longer. Note, however, that the same effect is done by adjusting $\lambda$ so that tuning one of both is enough in practice.

Note that a readout of a given matrix element of $\breve{A}$ happens every $n_s n$ input vectors (as the rows are sequentially read, see Fig. 1). Thus, after $t$ input vectors, $m = \lfloor \frac{t}{n_s n} \rfloor$ additions are made to the hidden matrix. Therefore, we set the learning rate $\lambda_H$ in Eq. (14) proportional with $\lambda_H \propto n_s n$ to avoid a weight update magnitude dependence on the potentially different layer sizes across the DNN.

To recover the original gradient magnitudes of the SGD that are written onto $\breve{W}$ approximately, the learning rate $\lambda_H$ in Eq. (14) is to be divided with $\lambda_A$, which scales the gradient accumulation onto $\breve{A}$ (however, note that we drop this dependence again for our empirical "Results" section, see paragraph "High-noise and high device asymmetry limit"). The value of $\lambda_A$ is dynamically adjusted. Since the conductance range is limited, the amount accumulated must be large enough to cause a significant change in the conductances of $\breve{A}$. We thus scale the learning rate $\lambda_A$ appropriately. Since the gradient magnitude often differs for individual layers, and might also change over time, we dynamically divide $\lambda_A$ by the recent running average $\mu_x$ and $\mu_d$ of the absolute maximum of the inputs $m_x = \max_j|x_j|$ and input gradients

$m_d = \max_i|d_i|$, respectively.

$$\lambda_A = \frac{\eta_0 l_{\max} \delta_{\breve{A}}}{\mu_x \mu_d} \qquad (15)$$

Note that $m_x$ and $m_d$ are needed for the gradient update already (see Supplementary Alg. 1), so that this does not require any additional computations, except for the scalar leaky average computations. Since $l_{\max}\delta_{\breve{A}}$ is approximately the maximum that the device material can change during one update ($l_{\max}$ is the number of pulses used, see Supplementary Alg. 1), the Eq. (15) means in case of $\eta_0 = 1$ that an element of the weight gradient is going to be clipped if $x_i d_j > \mu_x \mu_d$. The default value of $\eta_0$ is 1, although in some cases higher values improve learning.

**Expected weight update magnitude in limit cases.** It is instructive to investigate theoretically what weight update the algorithms are writing onto the weight matrix. Let's first assume an ideal device case without considering any feedback from the loss function in a typical gradient descent setting. Assume for simplicity that the gradient $\mathbf{dx}^T$ is constant for each $n$-dimensional input vector $\mathbf{x}$ and $n$-dimensional back-propagated error vector $\mathbf{d}$, that is $x_j d_i \equiv g$. Thus, after $t$ (identical) input vectors, the accumulated change of each weight element should be $\lambda g t$ (ignoring the sign of the descent), where $\lambda$ is the SGD learning rate.

Let's further assume that the learning rate $\lambda_A$ (see Eq. (15)) is roughly constant in the period of $t$ updates. According to the algorithms (see Fig. 1), each element $\breve{a}$ of $\breve{A}$ is read after a period of $n_s n$ input vectors and would then be $\breve{a}_{n_s n} = \lambda_A g n_s n$ in the ideal device case. Note that we write $\breve{a}_t$ for the value of $\breve{a}$ after $t$ input vectors. Since the algorithms will access each element of $\breve{A}$ $m = \lfloor \frac{t}{n_s n} \rfloor$ times and add the readout onto $H$, the value of the elements $h$ after $t$ input vectors are $h_t/\lambda_H = \sum_{i=1}^m \breve{a}_{i n_s n} = \sum_{i=1}^m i\, \breve{a}_{n_s n} = \lambda_A g n_s n \sum_{i=1}^m i$. Note that the term $c_m \equiv \sum_{i=1}^m i = \frac{(m+1)m}{2}$ results from the fact that (in the ideal case) the devices of $\breve{A}$ are not saturating or reset between reads.

Thus, we find $h_t = c_m \lambda_H \lambda_A g n_s n$. With Eq. (14) it is $h_t = c_m \frac{\lambda n_s n}{\gamma_0 \delta_{\breve{W}}} g n_s n$. Since $W$ is updated with $\delta_{\breve{W}}$ if $h > 1$ and $h$ is then reset according to the algorithms, we have $w_t = c_m \frac{\lambda n_s n}{\gamma_0} g n_s n$ and with $t \approx n_s n m \approx n_s n (m+1)$ it is $w_t \approx \lambda g t \frac{t}{2\gamma_0}$. Note that if one would set $\gamma_0 = \frac{t}{2}$ the SGD weight update amplitude is matched. For instance, $t$ could be the batch size (times re-use factor for convolutions).

**With choppers.** However, when we are using a chopper as in c-TTv2 and AGAD then the change of the chopper sign every $\frac{1}{\rho}$ readouts essentially resets the gradient accumulation on $\breve{A}$. If we correct (divide) the writing onto $H$ for the $k$-th read within a chopper cycle by $k$ then the pre-factor $c_m$ becomes just the number of reads in $t$ that is $m$. Thus, for the chopped algorithm (with multiple-read correction) it is $w_t \approx \lambda g t \frac{n_s n}{\gamma_0}$.

**High-noise and high device asymmetry limit.** In case of high device asymmetry and device noise, the accumulation on $\breve{A}$ quickly decays (with typical time constant of $\frac{1}{\delta_{\breve{A}}}$, see Eq. (7)). Thus, if the readout interval and device asymmetry is large, i.e., $n_s n \gg \frac{1}{\delta_{\breve{A}}}$, then the accumulated value is proportional to a filtered version of the instantaneous gradient $a_{n_s n} \propto \lambda_A \langle g \rangle$ with constant $c$ rather than proportional to $n_s n$ as in the ideal device case above. Thus, it is $c_m \approx m$, and the updated gradients are $w_t \approx \lambda \langle g \rangle t \frac{c}{\gamma_0}$. Therefore, the $n_s n$ dependence drops, which is the reason for the choice of Eq. (14).

In fact, it turns out empirically that the $\lambda_A$ dependence of Eq. (14), which re-scales the update on the weight with the incoming gradient magnitude, can be dropped as well. Effectively, the learning rate is then automatically normalized per layer based on the recent average

gradient magnitude ($\mu_x \mu_d$), since it is then $w_t \approx \lambda \langle g \rangle t \, \frac{c}{\gamma_0} \lambda_A \propto \frac{1}{\mu_x \mu_d}$ with Eq. (15). We find that this simplification works well in practice for our simulations where we assume noisy ReRAM-like devices (see "Results" section). However, we also confirmed that one can get similar accuracy results when adding the dependence of $\lambda_A$ as in Eq. (14) if the constant $\gamma_0$ is appropriately adjusted. The latter might be the preferred choice for larger or more heterogeneous DNNs to not alter the effective learning rate per layer and the overall dynamics of the learning in comparison to training in with standard FP SGD.

## Data availability

The training and test datasets used for this study are publicly available[34,37,52]. The raw data that support the findings of this study can be made available by the corresponding authors upon request after IBM management approval.

## Code availability

The full simulation code used for this study cannot be publicly released without IBM management approval and is restricted for export by the US Export Administration Regulations under Export Control Classification Number 3A001.a.9. However, the open source Apache License 2.0 (AIHWKIT) at https://github.com/IBM/aihwkit implements all algorithms discussed here[53].

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

## Acknowledgements
We thank the IBM Research AI HW Center and RPI for access to the AIMOS supercomputer, and the IBM Cognitive Compute Cluster for additional compute resources. We would like to thank Takashi Ando, Hsinyu (Sydney) Tsai, Nanbo Gong, Paul Solomon, and Vijay Narayanan for fruitful discussions.

## Author contributions
M.J.R. and T.G. conceived the study; M.J.R. conceived the AGAD algorithm, T.G. conceived the c-TTv2 algorithm. M.J.R. conducted all experiments and analyses, except the LSTM training experiments, done by F.C., the vision transformer experiments, done by O.I.F., and the MNIST-CNN experiments, done by M.J.R. and O.I.F.; M.J.R. wrote the manuscript.

## Competing interests
The authors declare no competing interests.
