## [Peer Review File · Nature Communications]

REVIEWER COMMENTS

Reviewer #1 (Remarks to the Author):

Summary:

This paper on Analog in-memory computations is a nice addition to the previous work by the authors (Ref 13 and Ref 15). Specifically, they have claimed that the new algorithms that they have developed (c-TTV2 and AGAD) building on top of the previous work for noisy analog systems scale with less computational penalty compared to the previous algorithms and are more efficient than corresponding digital computations.

Strengths:

The Analog in-memory as has been pointed out this paper and others (including the previous work by the authors) is computationally very efficient. This is very nicely highlighted in Table 1: The new algorithms introduced c-TTV2 and AGAD seem to scale with $O(N/ns)$, with impressive complexity for floating point ops. Specifically, AGAD which seems to be more significantly more efficient. The use of Kirchoff's and Ohm's laws of current conservation and conductance respectively to get $O(1)$ scaling from the previous papers (Refs 1-5) is worthwhile and significant and hence should be proven or demonstrated with real examples.

As this computational complexity should manifest in energy efficiency in simulations, these algorithms on the ensuing hardware (discussed with ReRAM and ECRAM devices) have the potential to be orders of magnitude lower energy than consumed by the conventional GPUs, CPUs, and the more widely used neuromorphic processors. This is an important result that would be good to be highlighted together with the suggested points discussed below.

The device material requirements including Endurance, Retention are discussed, are critical to the practical realization of specific computing as outlined in the paper.

Areas for improvements (general):

The paper should compare with other processors which are used for training (e.g. GPUs and other ASICs). Our suggestion is for them to refer to at least the following papers and make an effort for comparison of energy estimates in training, given their focus.

1. Zhou, Y., Dong, X., Akin, B., Tan, M., Peng, D., Meng, T., Yazdanbakhsh, A., Huang, D., Narayanaswami, R. and Laudon, J., 2021. Rethinking co-design of neural architectures and hardware accelerators. arXiv preprint arXiv:2102.08619

2. Shankar, S. and Reuther, A., 2022, September. Trends in Energy Estimates for Computing in AI/Machine Learning Accelerators, Supercomputers, and Compute-Intensive Applications. In 2022 IEEE High Performance Extreme Computing Conference (HPEC) (pp. 1-8). IEEE.

As indicated before, Analog in-memory discussed in this paper is building on previous work. Figure 5 shows the total error with number of states, but not with the number of epochs or iterations, or similar metrics. It will be helpful to the readers if the figures reflect the key conclusions of the paper.

Areas for improvements (specific):

Following are the specific suggestions for the authors.

1. Page 1 (“fundamental physics”) => “physical laws”. Here Kirchoff’s and Ohm’s laws are not considered fundamental physics unlike the laws of charge conservation on which the former is based. Ohm’s law is an approximation to current conductance under certain conditions.
2. Page 5, Sigma c-c is not described as used in equation (1).
3. What if symmetry points drift as iterations continue?
4. Page 19: “expected to be still orders of magnitude faster than alternatives”. Please show a specific example with these comparison results as indicated above. If not, can the authors try emulation?

Reviewer #2 (Remarks to the Author):

This paper describes two novel learning algorithms for training that can handle nonidealities in devices that current training methods ignore. Different training methods are covered in depth, such as TTV2, c-TTV2, and AGAD. Several devices are discussed such as capacitors, ReRAM, and ECRAM, and a generalized model is presented. This manuscript is well written and the novel training algorithms have significant potential. However, I believe that the device material model has some problems that need to be addressed before publication:

1. Section 2.3 – “Device material model” has several issues and needs to be improved. In particular Equation (1) is stated to be based on the model in [34, 35] but deviates significantly from the model

presented in [34, 35]. Modifications have been made to the model (which should be stated) and these modifications give rise to some issues discussed as follows:

a. It should be stated that the conductances are normalized and the normalization method should be explained upfront. (It might be helpful to use the variable w for the normalized weights as g implies that it is conductance).

b. As is, it appears that both of the conductance shifts are in the same direction because g_{\min} needs to be negative, which is not stated and isn't inferred until $\langle g_{\min} \rangle$ is stated to be -1. In addition, including g_{\max} and g_{\min} in the denominator is new to this manuscript and data is not included to justify this. None of the referenced papers include weight/conductance in the denominator. The $\langle g_{\text{zero}} \rangle$ equation is not quite correct, it should be the average rather than half of the range.

c. The introduction of the $\sigma \cdot \zeta$ parts to simulate randomness is reasonable but introduces edge cases that are not handled well. If g_{\max} or g_{\min} are 0 (a situation allowed by the max and min functions) then the deltas become infinite – which is not physically possible. In addition, several of the $\sigma \cdot \zeta$ terms can flip the sign of the delta resulting in either the positive shift being negative and the negative shift being positive or both shifts being in the same direction.

Overall, I recommend expanding this section and better explaining the fundamental model used (as well as matching it to data).

2. (Optional) It would improve the manuscript to perform simulation on datasets other than MNIST or at least discuss how these results could pertain to more difficult datasets. MNIST is not a particularly challenging dataset and is typically more tolerant of nonidealities than other datasets. It is possible that the c-TTv2 and AGAD algorithms perform much better than TTv2 on these other datasets. This could improve the potential impact of this manuscript.

3. (Minor) Please correct all grammatical errors. For example, in the abstract, “in-memory computing to accelerating the inference phase” should be “in-memory computing to accelerate the inference phase.”

Reviewer #3 (Remarks to the Author):

This paper proposes an end-to-end on-chip training algorithm for DNNs deployed on PCM crossbars by using device-aware cTTv2 and AGAD algorithms for backpropagation and weight update under asymmetric non-idealities. Experiments shown were comprehensive (in terms of hardware benchmarking of the proposed algorithms) and performed using the IBM AIHWKit for a realistic performance evaluation.

However, I have a few comments:

1. The measured accuracy results with the proposed algorithms were shown only for a simple MNIST task. I would suggest rerunning the accuracy results with more complex models/tasks such as Resnet-18/CIFAR10 and also report the associated hardware costs (as shown in Table 1).
2. I have a question regarding the transferability of the training methods if we change the underlying NVM device. As pointed out in Section 3.4, consider a case when a model is trained from scratch using one type of PCM device, does the accuracy transfer well to another device (say, PCM/RRAM/FeFET) fulfilling all requirements in Section 3.4? Is some amount of fine-tuning needed for a few epochs? Or do we again have to re-train the model fully? You can have experiments on this aspect to show the effectiveness of your proposed training algorithm. Apologies in case I missed something.
3. Finally, it will be good if the authors can comment on how their device aware training performs as compared to a lot of techniques that exist today [1, 3] in performing device noise aware training or batch norm adaptation post the actual weight training of NNs?
4. Related to the transferability comment, there are recent works [2] that talk about hybrid device NNs where certain layers are implemented on ReRAMs, certain layers on FeFET or PCM based on energy-latency-area requirements? Can the authors comment on if such hybrid device configuration can be supported by their cttv2 methodology and if such hybrid device configurations even make sense for on-device training?

The paper overall looks good. But, it will be great if the authors can comment on some of the high level points above and maybe bring it in a discussion section to help improve the paper.

[1] Bhattacharjee, Abhiroop, et al. "Examining the Role and Limits of Batchnorm Optimization to Mitigate Diverse Hardware-noise in In-memory Computing." arXiv preprint arXiv:2305.18416 (2023).

[2] Bhattacharjee, Abhiroop, Abhishek Moitra, and Priyadarshini Panda. "HyDe: A Hybrid PCM/FeFET/SRAM Device-search for Optimizing Area and Energy-efficiencies in Analog IMC Platforms." *IEEE Journal on Emerging and Selected Topics in Circuits and Systems* (2023).

[3] Meng, Jian, et al. "Temperature-resilient rram-based in-memory computing for dnn inference." *IEEE Micro* 42.1 (2021): 89-98.

REVIEWER COMMENTS

Reviewer #1 (Remarks to the Author)

Reviewer's comment:

Summary: This paper on Analog in-memory computations is a nice addition to the previous work by the authors (Ref 13 and Ref 15). Specifically, they have claimed that the new algorithms that they have developed (c-TTV2 and AGAD) building on top of the previous work for noisy analog systems scale with less computational penalty compared to the previous algorithms and are more efficient than corresponding digital computations.

Response: Many thanks for the accurate summary.

Reviewer's comment:

Strengths: The Analog in-memory as has been pointed out this paper and others (including the previous work by the authors) is computationally very efficient. This is very nicely highlighted in Table 1: The new algorithms introduced c-TTv2 and AGAD seem to scale with $O(N/ns)$, with impressive complexity for floating point ops. Specifically, AGAD which seems to be more significantly more efficient. The use of Kirchoff's and Ohm's laws of current conservation and conductance respectively to get $O(1)$ scaling from the previous papers (Refs 1-5) is worthwhile and significant and hence should be proven or demonstrated with real examples.

As this computational complexity should manifest in energy efficiency in simulations, these algorithms on the ensuing hardware (discussed with ReRAM and ECRAM devices) have the potential to be orders of magnitude lower energy than consumed by the conventional GPUs, CPUs, and the more widely used neuromorphic processors. This is an important result that would be good to be highlighted together with the suggested points discussed below.

The device material requirements including Endurance, Retention are discussed, are critical to the practical realization of specific computing as outlined in the paper.

Response: Many thanks for accurately highlighting our findings and for the suggestion to emphasize the benefits of the proposed algorithms in terms of energy efficient DNN training.

Reviewer's comment:

Areas for improvements (general): The paper should compare with other processors which are used for training (e.g. GPUs and other ASICs). Our suggestion is for them to refer to at least the following papers and make an effort for comparison of energy estimates in training, given their focus.

1. Zhou, Y., Dong, X., Akin, B., Tan, M., Peng, D., Meng, T., Yazdanbakhsh, A., Huang, D., Narayanaswami, R. and Laudon, J., 2021. Rethinking co-design of neural architectures and hardware accelerators. *arXiv preprint arXiv:2102.08619*
2. Shankar, S. and Reuther, A., 2022, September. Trends in Energy Estimates for Computing in AI/Machine Learning Accelerators, Supercomputers, and Compute-Intensive Applications. In *2022 IEEE High Performance Extreme Computing Conference (HPEC) (pp. 1-8)*. IEEE.

Response: Many thanks for these important references. We have now added the following paragraphs discussing energy efficiency and the suggested references. The first paragraph reads:

We like to emphasize that the reduced number of digital operations necessary for our AIMC training algorithm, together with the non-von-Neumann architecture and high energy efficiency of MVMs and outer products on the analog crossbar arrays, translates into an highly energy efficient approach for DNN training in comparison to traditional digital ways of compute. While the energy efficiency per digital operation has improved over time [Shankar et al. 2022], the complexity of computing MVMs remains still $\mathcal{O}(N^2)$ on digital and is thus inherently worse than our AIMC approach. While we here have given a runtime estimate for the gradient update only, a full energy estimate for a complete DNN training run on a AIMC architecture is beyond this study. Such an estimate has to take into account not only the forward computations of linear layers, auxiliary digital compute such as normalization and activation functions, pipelining of parallel crossbar arrays, etc., as recently shown for an energy estimate for inference only AIMC hardware [Jain et al. 2022], but also the backward pass and the update computations as well as the storing of intermediate results (see [Gokmen 2021] for a discussion).

The second added paragraph discusses the idea of the second reference suggested by the reviewer, which we think is indeed very important, namely the opportunity to co-design DNN architecture for AIMC:

Indeed, even more energy savings could result from co-designing DNNs for deployment on AIMC architectures, as the scaling laws of the SGD training compute is different compared to digital hardware. For instance, a large and dense matrix-multiplication is much less costly on AIMC than on digital von-Neumann hardware, potential opening up opportunities for designing novel energy efficient DNN architectures with high accuracy tailored to AIMC in the spirit of [Zhou et al 2021].

Reviewer’s comment:

As indicated before, Analog in-memory discussed in this paper is building on previous work. Figure 5 shows the total error with number of states, but not with the number of epochs or iterations, or similar metrics. It will be helpful to the readers if the figures reflect the key conclusions of the paper.

Response: We thank the reviewer for this good suggestion. Indeed, it is important to confirm the implicit assumption that training in-memory does not significantly alter the number of epochs needed for convergence. We have now added example plots showing the test error as a function of training epochs that confirms this assumption (see Supplementary Figures 5-6). Note that in Figure 5 of the main text, we use the same number of epochs for all simulations, so that results are comparable.

However, to emphasize this important point we also added the following paragraph in Section 3.5 ‘Performance’:

[..] we assume that a similar amount of training epochs are needed to reach acceptable accuracy. We confirmed that the number of epochs needed for convergence is indeed on the same order of magnitude in respect to the FP baseline

in practice (see Supplementary Fig. A.5 for example traces for the data in Fig. 5 A), validating our assumption in first-order approximation.

Reviewer’s comment:

Areas for improvements (specific): Following are the specific suggestions for the authors. 1. Page 1 (“fundamental physics”) = \hat{c} “physical laws”. Here Kirchoff’s and Ohm’s laws are not considered fundamental physics unlike the laws of charge conservation on which the former is based. Ohm’s law is an approximation to current conductance under certain conditions.

Response: We agree and have changed the term to “basic physical laws of electrostatics”.

Reviewer’s comment:

2. Page 5, Sigma c-c is not described as used in equation (1).

Response: We thank the reviewer pointing us at this issue. Indeed, the incremental update noise was not correct in the earlier version. We actually used additive noise in the simulation as it also fits experimental observation with ReRAM devices better [Gong et al. IEDM 2023], but this was not reflected in the equation of our initial submission. We have now corrected the equations. We also enlarged the description of the device model to make the parameters clearer (see Sec. 2.3 (“Device Material Model”)).

Reviewer’s comment:

3. What if symmetry points drift as iterations continue?

Response: This is one of the potential error sources we are discussing in Figure 4-5 (reference offset variation). While we here add error to the stored value (assuming that the symmetry point does not change), it is almost identical to the case when the symmetry point itself drifts and the stored value remains the same. As shown in the plots, if the stored estimate of the symmetry point does not accurately reflect the actual symmetry point of the device, the algorithms TTV2 and c-TTV2 will diverge. However, AGAD is invariant against such a change, as the reference point is calculated and updated on-the-fly (assuming that the change occurs on a much slower time scale than the duration of a chopper cycle).

Reviewer’s comment:

4. Page 19: “expected to be still orders of magnitude faster than alternatives”. Please show a specific example with these comparison results as indicated above. If not, can the authors try emulation?

Response: This statement is solely based on the projected average gradient update runtime per input data. This comparison thus assumes that the convergence speed with similar learning rate is not significantly impacted. As mentioned above, we have now included more details on the convergence in the experiments to validate this point (see Supplementary Figures 5-6).

To make the statement clearer, we added the sentence:

Indeed, when estimating the average gradient update time for a 512×512 weight matrix in Tab.1 with reasonable assumptions, we find 62.1 ns for AGAD versus > 3000 ns when updating the gradient matrix in digital instead.

Reviewer #2 (Remarks to the Author):

Reviewer's comment:

This paper describes two novel learning algorithms for training that can handle nonidealities in devices that current training methods ignore. Different training methods are covered in depth, such as TTv2, c-TTv2, and AGAD. Several devices are discussed such as capacitors, ReRAM, and ECRAM, and a generalized model is presented. This manuscript is well written and the novel training algorithms have significant potential.

Response: We thank the reviewer for the accurate summary and appreciate the generally positive evaluation.

Reviewer's comment:

However, I believe that the device material model has some problems that need to be addressed before publication:

- 1. Section 2.3 – “Device material model” has several issues and needs to be improved. In particular Equation (1) is stated to be based on the model in [34, 35] but deviates significantly from the model presented in [34, 35].*

Response: We thank the reviewer for pointing us to this section, as it was indeed not clear. In fact, the soft bounds model proposed by [34] is much simpler compared to ours, as it does not consider any device-to-device variation or additional noise sources. However, when these device-to-device variations and noise sources are turned off, our model is equivalent to that of [34] up to a different normalization of the conductance ([34] normalizes the conductances in the range 0 to 1, whereas we normalize it into the range -1 to 1). We have now expanded the section significantly, deriving our model in more detail, to make the discrepancies and similarities more clear. Please refer to Sec. 2.3. (“Device material model”) in the revision.

Reviewer's comment:

Modifications have been made to the model (which should be stated) and these modifications give rise to some issues discussed as follows:

- a. It should be stated that the conductances are normalized and the normalization method should be explained upfront. (It might be helpful to use the variable w for the normalized weights as g implies that it is conductance).*

Response: Indeed, the conductance are given in normalized units, which is, however, standard practice, as also [34] normalizes the conductances for easier discussions. Note, however, that the form of normalization does not effect the results in anyway as it is a simple unit change. We have made our normalization now clearer in the rewriting of the section and write \tilde{w} for the normalized conductance values instead of previously g to not confuse the reader.

Reviewer's comment:

- b. As is, it appears that both of the conductance shifts are in the same direction because g_{min} needs to be negative, which is not stated and isn't inferred until $\langle g_{min} \rangle$ is stated to be -1. In addition, including g_{max} and g_{min} in the denominator is new to this manuscript and data is not included to justify this.*

None of the referenced papers include weight/conductance in the denominator. The $\langle g_{\text{zero}} \rangle$ equation is not quite correct, it should be the average rather than half of the range.

Response: Admittedly, we did not explicitly state that our model extends the original soft-bound of Fusi & Abbott significantly by adding cycle-to-cycle noise as well as device-to-device variations on the saturation levels and asymmetry. However, if these additions are turned off, the models are identical up to a different normalization of the conductances.

We have now changed the section 2.3 to better discuss the differences. In detail, the section now reads:

[.] According to that model, the conductance change $g \leftarrow g + \Delta g_D$ to a single voltage pulse in either up ($D = +$ or down $D = -$) direction is given by

$$\begin{aligned}\Delta g_+ &\equiv \alpha_+ (g_{\max} - g) \\ \Delta g_- &\equiv \alpha_- (g_{\min} - g)\end{aligned}\tag{1}$$

where thus the induced conductance change gradually reduces towards the conductance bounds. While here the conductance is measured in physical units, it is more convenient for the following discussion to (arbitrarily) normalize the conductances. For that, we first set $\langle g_{\text{center}} \rangle \equiv \frac{\langle g_{\max} \rangle - \langle g_{\min} \rangle}{2}$ where the average is taken over the individual devices (that in general have individual g_{\min} and g_{\max} values due to device-to-device variations). Then, we set the normalized conductance value to $\check{w} \equiv \frac{g - \langle g_{\min} \rangle}{\langle g_{\text{center}} \rangle} - 1$, so that for a device at $\langle g_{\min} \rangle$ the normalized value is $\check{w} = -1$, and $\langle g_{\max} \rangle$ corresponds to $\check{w} = 1$, and finally $\langle g_{\text{center}} \rangle$ corresponds to $\check{w} = 0$.

Using this normalization, Eq. 1 becomes (assuming no device variations for the moment, i.e. $g_{\max} = \langle g_{\max} \rangle$ and $g_{\min} = \langle g_{\min} \rangle$)

$$\begin{aligned}\Delta \check{w}_+ &\equiv \alpha_+ (1 - \check{w}) \\ \Delta \check{w}_- &\equiv -\alpha_- (1 + \check{w})\end{aligned}\tag{2}$$

which corresponds to the soft-bounds model in [Fusi & Abbott 2007] albeit with a different conductance normalization (here shifted to the range of $-1, \dots, 1$ instead of $0, \dots, 1$ to ease of discussion of the algorithmic zero point).

We introduce device-to-device variations on the saturation levels as well as on the slope parameter α and cycle-to-cycle update fluctuations to arrive at the full model

$$\begin{aligned}\Delta \check{w}_+ (\check{w} | \boldsymbol{\theta}) &\equiv \alpha_+ \left(\frac{\check{w}_{\max} - \check{w}}{\check{w}_{\max}} + \sigma_{\text{c-to-c}} \xi \right) \\ \Delta \check{w}_- (\check{w} | \boldsymbol{\theta}) &\equiv -\alpha_- \left(\frac{\check{w}_{\min} - \check{w}}{\check{w}_{\min}} + \sigma_{\text{c-to-c}} \xi \right)\end{aligned}\tag{3}$$

where ξ are standard normal random numbers (drawn for each update) to model the update fluctuations of strength $\sigma_{\text{c-to-c}}$. Here we chose to normalize the difference of the actual conductance to the bound by the bound, that is e.g. $\frac{\check{w}_{\max} - \check{w}}{\check{w}_{\max}}$, so that the update size remains constant for the same relative distance of \check{w} towards the bounds when varying solely \check{w}_{\min} or \check{w}_{\max} . Note that in simulations the normalized conductance values are ensured to be clamped

to the saturation levels (between \check{w}_{min} and \check{w}_{max}) to avoid that the additive noise would drive the conductance to not supported levels.

[..].

Please also refer to our redlined version of the revision to see all changes in the text.

Reviewer’s comment:

*c. The introduction of the sigma*zeta parts to simulate randomness is reasonable but introduces edge cases that are not handled well. If g_max or g_min are 0 (a situation allowed by the max and min functions) then the deltas become infinite – which is not physically possible. In addition, several of the sigma*zeta terms can flip the sign of the delta resulting in either the positive shift being negative and the negative shift being positive or both shifts being in the same direction.*

Response: That is correct, we therefore force the conductance to the bounds after each update, which was not explicitly stated in the text. We have added it in the revision. We now write:

Note that in simulations the normalized conductance values are ensured to be clamped to the saturation levels (between \check{w}_{min} and \check{w}_{max}) to avoid that the additive noise would drive the conductance to not supported levels.

Reviewer’s comment:

Overall, I recommend expanding this section and better explaining the fundamental model used (as well as matching it to data).

Response: We thank the reviewer for the good suggestion. We have now considerably expanded this section (see above), which should make the material device model much clearer.

Reviewer’s comment:

2. (Optional) It would improve the manuscript to perform simulation on datasets other than MNIST or at least discuss how these results could pertain to more difficult datasets. MNIST is not a particularly challenging dataset and is typically more tolerant of nonidealities than other datasets. It is possible that the c-TTv2 and AGAD algorithms perform much better than TTv2 on these other datasets. This could improve the potential impact of this manuscript.

Response: We agree that MNIST is quite small, however, it is a typically used benchmark dataset in previous studies. Note that apart from a fully connected network and a convnet on MNIST, we had already included a recurrent network (LSTM) on a different data set (text classification on the War and Peace novel), which is more challenging than image classification on MNIST. We showed that the results are very well transferable. In general, we find that TTv2 performs very competitively for cases where no noise is added to the symmetry point reference. However, if realistic noise is added c-TTv2 and AGAD is vastly superior.

In this revision, however, we added another larger model as suggested by the reviewer. We now confirmed the general trend also on a Vision transformer on CIFAR10 for image classification, having more than 10× more parameters (4.3M).

The added paragraph on this vision transformer experiment now reads:

Although these three benchmark networks have been used extensively in previous studies on AIMC training algorithm evaluation, they are relatively small in terms of free parameter (235K, 80K, and 77K, respectively). We thus further trained a vision transformer [Lee et al 2021] for classifying the CIFAR10 [Krizhevsky and Hinton 2009] image data set, which is significantly larger (4.3M parameter; see Supplementary Methods Sec. C.1.4 for details). Consistently with the results on the benchmark DNNs, we found that when no noise is added to the symmetry point reference, the classification error is 36.1% (0.6% SD over 5 trials), 35.9% (0.4% SD), and 37.5% (0.6% SD) for TTv2, c-TTv2, and AGAD, respectively, however, if a SD of $\sigma_r = 0.25$ is added (compare to Fig. 5 B-D), then test errors become 59% (SD 0.1%), 37.9% (SD 0.4%), and 37.3% (SD 0.3%), respectively. This again shows the susceptibility of TTv2 for an inaccurate symmetry point reference value, in contrast to the proposed algorithmic improvements c-TTv2 and AGAD. Here, $n_{states} = 200$ are assumed and the FP test error is 29.3% (without image augmentation). Note that due to simulation time limitations for this relatively large DNN, no hyper-parameter optimization was performed in this experiment, which might explain the remaining gap to the FP accuracy and thus resulting accuracy could potentially be further improved.

Reviewer’s comment:

3. (Minor) Please correct all grammatical errors. For example, in the abstract, “in-memory computing to accelerating the inference phase” should be “in-memory computing to accelerate the inference phase.”

Response: Many thanks for this suggestion. We have now corrected a number of such typo.

Reviewer #3 (Remarks to the Author):

Reviewer’s comment:

This paper proposes an end-to-end on-chip training algorithm for DNNs deployed on PCM crossbars by using device-aware cTTv2 and AGAD algorithms for backpropagation and weight update under asymmetric non-idealities. Experiments shown were comprehensive (in terms of hardware benchmarking of the proposed algorithms) and performed using the IBM AIHWKit for a realistic performance evaluation.

Response: We thank the reviewer for this summary. Please note, however, that we are not proposing the use of PCM (phase change materials) as device material here, as it would not support the required bi-directional incremental updates. Instead our new algorithms is designed for other device materials for instance ECRAM, ReRAM, or capacitance CMOS, or any other bi-directionally switching material.

Reviewer’s comment:

However, I have a few comments:

1. *The measured accuracy results with the proposed algorithms were shown only for a simple MNIST task. I would suggest rerunning the accuracy results with more complex models/tasks such as Resnet-18/CIFAR10*

Response: We agree that confirming the feasibility of the algorithms on larger networks is desirable. However, note again that we here simulate the gradient update itself in-memory with stochastic pulsed conductance update and are not considering the more common hardware-aware training approach (for AIMC inference chip deployment) that uses perfect (floating point) update and backward pass processing instead. Therefore, the update pass uses non-linear device materials as well as explicit simulation of each of the stochastic pulses. The simulation is thus naturally much more demanding than hardware-aware training for inference chips.

Furthermore, while we have used MNIST as a benchmark dataset for two of the DNNs to make an easier comparison to previous studies that used the same networks and datasets, we actually had already included a recurrent network DNN and task (text classification on the War and Peace novel) that is indeed considerably more complex than MNIST digit classification (see Figure 5D). We showed that results remain qualitatively similar.

Still, we acknowledge the reviewer’s concern and have now added another experiment using a vision transformer for image classification (CIFAR10). Note that the convolution architecture is not particularly well suited for in-memory AIMC architectures without any adaptations as they have a high and unequal weight reuse factor between layers and often small kernel matrices favoring the compute model of GPUs (see Rasch et al. Front. in Neurosci. 2009 for a discussion). However, more modern transformer architectures that require in general large and dense matrix computation are a better fit (see also Jain et al. TVLSI 2023 for a discussion).

We found that the results on this larger DNN (4.3M parameter) are very consistent with those obtained on the smaller benchmark DNNs. The added paragraph on these new results reads:

Although these three benchmark networks have been used extensively in previous studies on AIMC training algorithm evaluation, they are relatively small in terms of free parameter (235K, 80K, and 77K, respectively). We thus further trained a vision transformer [Lee et al 2021] for classifying the CIFAR10 [Krizhevsky and Hinton 2009] image data set, which is significantly larger (4.3M parameter; see Supplementary Methods Sec. C.1.4 for details). Consistently with the results on the benchmark DNNs, we found that when no noise is added to the symmetry point reference, the classification error is 36.1% (0.6% SD over 5 trials), 35.9% (0.4% SD), and 37.5% (0.6% SD) for TTv2, c-TTv2, and AGAD, respectively, however, if a SD of $\sigma_r = 0.25$ is added (compare to Fig. 5 B-D), then test errors become 59% (SD 0.1%), 37.9% (SD 0.4%), and 37.3% (SD 0.3%), respectively (see also Supplementary Fig. 6). This again shows the susceptibility of TTv2 for an inaccurate symmetry point reference value, in contrast to the proposed algorithmic improvements c-TTv2 and AGAD. Here, $n_{states} = 200$ are assumed and the FP test error is 29.3% (without image augmentation). Note that due to simulation time limitations for this relatively large DNN, no hyper-parameter optimization was performed in this experiment, which might explain the remaining gap to the FP accuracy and thus resulting accuracy could potentially be further improved.

Reviewer’s comment:

and also report the associated hardware costs (as shown in Table 1).

Response: Note that the cost in Table 1 are only the cost for purely performing one single MVM update and are thus agnostic to any DNN architecture. To arrive at the hardware cost for a full DNN architecture, many more aspects need to be considered, such as intermediately storing results of the forward pass, the depth of the network and so on and is thus beyond this study. We have now made this point clearer in the discussion, where we now write:

We here have given a runtime estimate for the gradient update only, instead of a complete estimate of training the DNN on a given chip. A complete estimate has to take into account many details of the mixed analog-digital chip architecture, as it needs to consider not only the forward pass computations of all analog and digital auxiliary layers (as recently shown for an energy estimate for inference only AIMC hardware [Jain et al. 2022]), but also the backward pass, and weight update computations that require intermediately storing of results (see [Gokmen 2021] for a discussion). Therefore, a complete energy estimate for a full DNN training run has to be based on a specific AIMC chip architecture and is thus beyond the current study.

Reviewer’s comment:

2. I have a question regarding the transferability of the training methods if we change the underlying NVM device. As pointed out in Section 3.4, consider a case when a model is trained from scratch using one type of PCM device, does the accuracy transfer well to another device (say, PCM/RRAM/FeFET) fulfilling all requirements in Section 3.4? Is some amount of fine-tuning needed for a few epochs? Or do we again have to re-train the model fully? You can have experiments on this aspect to show the effectiveness of your proposed training algorithm. Apologies in case I missed something.

Response: We here describe algorithms that consider updating the conductances of a crossbar arrays in-place and thus implement the full SGD training process. Therefore, the NVM devices are assumed to have random values at the beginning. In contrast, what the reviewer describes is similar to the application of fine-tuning of previously learned weights by other means. Instead, we assume in our study that inference is done with the same devices the training was done without the need to program any conductances (e.g. for online learning in edge applications). Indeed, the SGD training itself can be thought of as a way to program the conductances to suitable values. That being said, in principle the weights can also be extracted (read out) from the conductances and then deployed on another hardware instance. This approach is described in more detail in [Gokmen 2021], where stochastic weight averaging is suggested as a promising way to read out the weights from the conductances. This method, suggested for TTv2, could similarly be applied for c-TTv2 and AGAD, since these algorithms only differ in terms of how the gradient update is computed in-memory but not in the way the weight matrix is represented by conductances in the forward pass.

However, in principle one could also try to program a previously trained weight matrix into the conductances and then fine-tune the weights further using the proposed algorithms, similar to the approach of transfer-learning applications. In this case, the programming of weights into the conductances will introduce a weight representation error, however, due to the noisy update behaviour. In fact, Fig. 4. exactly discusses this weight programming error depending on the device model and the accuracy of the symmetry point estimation using the proposed algorithms. For instance, in case of AGAD,

the weight programming error can remain very low in many cases, depending on the precision of the device itself. Such an error will typically translate to a drop in inference accuracy depending on DNN and application. How much exactly would be an interesting future topic, however, we feel that this transfer-learning approach is beyond this paper to investigate.

Reviewer’s comment:

3. Finally, it will be good if the authors can comment on how their device aware training performs as compared to a lot of techniques that exist today [1, 3] in performing device noise aware training or batch norm adaptation post the actual weight training of NNs?

Response: We thank the reviewer for this suggestions. Indeed, it is important to clarify the difference of our approach from the device-aware training methods proposed by e.g. in the papers [1, 3]. In device-aware training, the simulation is done in software and typically some noise properties are included in the forward pass with the goal to making the DNN converge to a more noise robust solution. The resulting weights are then simply programmed onto an AIMC hardware for inference applications. This is completely different to what we are proposing here. Here, we are not proposing a new device-aware training method in software, but instead propose a viable way to do the training of the DNN directly on the AIMC chip with great runtime benefits. So our target application is a DNN *training* accelerator hardware, whereas the cited studies consider a DNN *inference* accelerator hardware instead. Note that in the latter case, additionally a software training has to be performed to get the converged weights that are then simply programmed onto the conductances.

We also like to emphasize that our approach of accelerating DNN training using AIMC is much more challenging than merely accelerating inference, since not only the forward pass is affected by device noise, but also backward and gradient is affected by non-linear device properties and noise (see also Supplementary Fig. 1).

After training the DNN on the AIMC chip, in principle an additional inference application could be considered. In our approach, the inference could either be done using the same chip (without the need to extract the weights, e.g. for online training of edge devices) or one could use methods to extract the weight to a high degree of fidelity (using stochastic weight averaging) and use other hardware for inference altogether, as discussed in more detail in [Gokmen 2021].

We have added a paragraph to the discussion to clarify these aspects. We now write:

Note that this in-memory training approach is radically different from the hardware-aware training typically employed when using analog crossbar arrays for DNN inference only (e.g. [Rasch et al 2023, Bhattacharjee et al 2023, Meng et al 2022]). In the latter case, the DNNs weights are (re)-trained in software (using traditional digital CPUs or GPUs) assuming generic noise sources to improve the noise robustness of the weights, and only then programmed onto the analog AI hardware accelerator before an inference application. In contrast, in our study the training of the weights itself is done by the analog AI hardware accelerator in-memory on the crossbar arrays, thus opening up the possibility for high energy efficiency during the training of DNNs. Whether inference is then done with the same hardware using the trained weights depends on the application. While directly using the trained weights with the same hardware for inference would be the most efficient, other

approaches are possible as well. For instance, Gokmen [Gokmen 2021] suggests to extract the trained weights during in-memory training using stochastic weight averaging in a highly efficient way, so that they can then be used for any other hardware during training, including reduced precision digital inference accelerators. Other analog inference hardware could be used as well, however, an additional programming error penalty will be introduced in this case. Nevertheless, given that realistic device noise is naturally present during our proposed in-memory training, the resulting weights are likely to be robust to any device noise in a way similar to the conventional hardware-aware training approach in software (see e.g. [Rasch et al]).

Reviewer's comment:

4. Related to the transferability comment, there are recent works [2] that talk about hybrid device NNs where certain layers are implemented on ReRAMs, certain layers on FeFET or PCM based on energy-latency-area requirements? Can the authors comment on if such hybrid device configuration can be supported by their ctv2 methodology and if such hybrid device configurations even make sense for on-device training?

Response: While the work suggested by the reviewer raises an interesting idea to optimize the device materials based on the DNN needs, we here typically consider that the same materials for all layers, to increase general applicability of the hardware (to any DNN and not a specific DNN topology). In principle, however, different layers might indeed need different hyper-parameter settings, and possibly it would be possible to consider different materials for different layers, for instance the input and output often require high precision and thus favor less device noise. The choice of the device material is, however, is restricted to bi-directionally switching material, therefore PCM is not supported by in-device training. Moreover, as we discuss in Figure 6, asymmetry in the device switching is further needed for c-TTv2 (but not for AGAD).

Reviewer's comment:

The paper overall looks good. But, it will be great if the authors can comment on some of the high level points above and maybe bring it in a discussion section to help improve the paper.

[1] Bhattacharjee, Abhiroop, et al. "Examining the Role and Limits of Batch-norm Optimization to Mitigate Diverse Hardware-noise in In-memory Computing." *arXiv preprint arXiv:2305.18416* (2023).

[2] Bhattacharjee, Abhiroop, Abhishek Moitra, and Priyadarshini Panda. "HyDe: A Hybrid PCM/FeFET/SRAM Device-search for Optimizing Area and Energy-efficiencies in Analog IMC Platforms." *IEEE Journal on Emerging and Selected Topics in Circuits and Systems* (2023).

[3] Meng, Jian, et al. "Temperature-resilient rram-based in-memory computing for dnn inference." *IEEE Micro* 42.1 (2021): 89-98.

Response: We thank the review for the suggestions for improvements.

REVIEWERS' COMMENTS

Reviewer #2 (Remarks to the Author):

The authors have significantly improved the manuscript and its potential impact with the improved explanation of the model and additional dataset. I have two optional revision/recommendations to consider.

I recommend reexamining the equations for g_center between eq1 and eq2. The variable defined as g_center in $g_center = (g_max - g_min)/2$ and $w = (g - g_min)/g_center - 1$ is better defined as $g_halfrange$, or something to that effect, for the following reason. In the last sentence of the paragraph it is stated that g_center corresponds to $w=0$ which, according to the last two equations, is not correct. The value of g that corresponds to $w=0$ is $(g_max + g_min)/2$. This could be fixed in a number of ways, but as it currently stands it is not quite correct.

Second, the CIFAR-10 error is quite high by modern standards, and even the FP error is high compared to the referenced recipe that was used. To some extent this is partially mitigated by the fact that the FP version and analog versions all have high error, and the purpose is mainly to check the comparative accuracy trends are being included (and this is only included in the supplement). However, the authors should consider better addressing this issue, as the readers will almost certainly question this. This also could lead to confusion that AIMC training has fundamental accuracy problems.

Reviewer #3 (Remarks to the Author):

All comments have been addressed.

REVIEWER COMMENTS

Reviewer #2 (Remarks to the Author)

Reviewer's comment:

The authors have significantly improved the manuscript and its potential impact with the improved explanation of the model and additional dataset.

Response: We thank the reviewer very much for the constructive feedback and great suggestions for improvement.

Reviewer's comment:

I have two optional revision/recommendations to consider.

I recommend reexamining the equations for g_{center} between eq1 and eq2. The variable defined as g_{center} in $g_{center} = (g_{max} - g_{min})/2$ and $w = (g - g_{min})/g_{center} - 1$ is better defined as $g_{half-range}$, or something to that effect, for the following reason. In the last sentence of the paragraph it is stated that g_{center} corresponds to $w = 0$ which, according to the last two equations, is not correct. The value of g that corresponds to $w = 0$ is $(g_{max} + g_{min})/2$. This could be fixed in a number of ways, but as it currently stands it is not quite correct.

Response: We thank the reviewer for pointing us to the mistake as $\check{w} = 0$ indeed actually corresponds to $\frac{\langle g_{min} \rangle + \langle g_{max} \rangle}{2}$ and not $\frac{\langle g_{max} \rangle - \langle g_{min} \rangle}{2}$ as we wrongly stated earlier. We corrected this in the revision. This, however, can be considered rather a typo in explaining the normalization as it has no effect on the following analysis. We also agree with the first suggestion of the reviewer and renamed g_{center} to $g_{half-range}$ as this name is indeed clearer.

Reviewer's comment:

Second, the CIFAR-10 error is quite high by modern standards, and even the FP error is high compared to the referenced recipe that was used. To some extent this is partially mitigated by the fact that the FP version and analog versions all have high error, and the purpose is mainly to check the comparative accuracy trends are being included (and this is only included in the supplement). However, the authors should consider better addressing this issue, as the readers will almost certainly question this. This also could lead to confusion that AIMC training has fundamental accuracy problems.

Response: We thank the reviewer for this valid feedback and suggestion. The CIFAR-10 results using the larger vision transformer DNN are here indeed presented to show a consistent trend and are not optimized for giving the highest possible accuracy, as this would require some more tuning of the hyper-parameters and simulation time. For instance, when image augmentation is included additionally, typical reported training use 500 epochs (see <https://github.com/kentaroy47/vision-transformers-cifar10>) compared to our run with 40 epochs.

However, since the raw accuracy in this experiment is not central to our conclusion – but rather the trend among the algorithms when offsets are present – and we do not want to invite any misunderstandings, we have now rephrased the discussion of the vision transformer experiment in more cautious terms and put the main results of this experiments into the appendix.

We now write in the main text:

Although these three benchmark networks have been used extensively in previous studies on AIMC training algorithm evaluation, they are relatively small in terms of free parameter (235K, 80K, and 77K, respectively). Simulating every update pulse for each weight element accurately in larger networks remains challenging due to simulation time limitations, in particular when multiple training runs are necessary for hyper-parameter tuning. However, to confirm whether the general trend of the effect of a reference value offset on the various algorithms is preserved in larger DNNs, we conducted a brief training experiment on a vision transformer [42] for classifying the CIFAR-10 [43] image data set, which is significantly larger (4.3M parameter; see Supplementary Methods Sec. C.1.4 for details). Indeed, even without hyper-parameter tuning, we found that when the reference offset is not perfectly corrected for, the classification error remains markedly stable only for the proposed algorithmic improvements c-TTv2 and AGAD but not for TTv2 (see Supplementary Methods Sec. C.1.4). This is very consistent with the observed trend for the smaller benchmark DNNs (compare to Fig. 5 B-D).

In the Supplementary Information (C.1.4) we added the following paragraph:

Consistently with the results on the benchmark DNNs, we found that when no noise is added to the symmetry point reference, the classification error is 36.1% (0.6% SD over 5 trials), 35.9% (0.4% SD), and 37.5% (0.6% SD) for TTv2, c-TTv2, and AGAD, respectively, however, if a SD of $\sigma_r = 0.25$ is added (compare to Fig.5 B-D), then test errors become 59% (SD 0.1%), 37.9% (SD 0.4%), and 37.3% (SD 0.3%), respectively (see also SI Fig A.6). This again shows the susceptibility of TTv2 for an inaccurate symmetry point reference value, in contrast to the proposed algorithmic improvements c-TTv2 and AGAD.

Here, the FP test error is 29.3% using our setup (without image augmentation, 40 epochs, no transfer learning). Note again that due to simulation time limitations for this relatively large DNN, no hyper-parameter optimization was performed for the analog training algorithms for this experiment, which might explain the remaining gap to the FP accuracy. Also, image augmentation with longer training runs, as well as transfer learning, is likely to improve the obtainable accuracy considerably, which, however, is beyond the focus of the current study.

Reviewer #2 (Remarks to the Author)

Reviewer's comment:

All comments have been addressed.

Response: We thank the reviewer very much for the time and effort reading our manuscript.